# Physical interactions between specifically regulated subpopulations of the MCM and RNR complexes prevent genetic instability

Aurora Yáñez-Vilches[1], Antonia M. Romero[1], Marta Barrientos-Moreno[1], Esther Cruz[1], Román González-Prieto[1,2], Sushma Sharma[3], Alfred C. O. Vertegaal[2], Félix Prado[1]*

1 Centro Andaluz de Biología Molecular y Medicina Regenerativa–CABIMER, Consejo Superior de Investigaciones Científicas, Universidad de Sevilla, Universidad Pablo de Olavide, Seville, Spain,
2 Department of Cell and Chemical Biology, Leiden University Medical Center, Leiden, the Netherlands,
3 Department of Medical Biochemistry and Biophysics, Umeå University, Umeå, Sweden

* felix.prado@cabimer.es

**Data Availability Statement:** All relevant data are within the manuscript and its Supporting Information files. Raw data are shown in S9 Fig

## Abstract

The helicase MCM and the ribonucleotide reductase RNR are the complexes that provide the substrates (ssDNA templates and dNTPs, respectively) for DNA replication. Here, we demonstrate that MCM interacts physically with RNR and some of its regulators, including the kinase Dun1. These physical interactions encompass small subpopulations of MCM and RNR, are independent of the major subcellular locations of these two complexes, augment in response to DNA damage and, in the case of the Rnr2 and Rnr4 subunits of RNR, depend on Dun1. Partial disruption of the MCM/RNR interactions impairs the release of Rad52 –but not RPA–from the DNA repair centers despite the lesions are repaired, a phenotype that is associated with hypermutagenesis but not with alterations in the levels of dNTPs. These results suggest that a specifically regulated pool of MCM and RNR complexes plays non-canonical roles in genetic stability preventing persistent Rad52 centers and hypermutagenesis.

## Author summary

Cells have to accurately replicate their genomes to avoid loss of genetic information during cell divisions. This is achieved by coordinating the advance of the replication machinery with different mechanisms that prevent and/or repair errors during DNA synthesis. However, different DNA lesions accumulate during DNA replication that are repaired at DNA repair centers by error-free homologous recombination and error-prone translesion synthesis mechanisms. Here, we show that the replicative helicase and the ribonucleotide reductase complexes that provide the substrates (ssDNA templates and dNTPs, respectively) for DNA replication interact with each other in a DNA damage-regulated manner. Furthermore, we observe that these interactions facilitate the dynamics of a key recombination protein (Rad52) at the repair centers and prevent genetic instability by translesion synthesis through mechanisms that are not associated with the canonical functions of the

and S3 Table. The mass spectrometry data have been deposited to the ProteomeXchange Consortium via the PRIDE partner repository with the data set identifier PXD045914.

**Funding:** This work was supported by grants PGC2018-099182-B-100 and PID2021-127486NB-100, funded by MCIN/AEI/10.13039/501100011033 and "ERDF A way of making Europe" (to FP) and ERC 310913, funded by the European Research Council (to ACOV). AY-V, AM.R and SS were recipients of a pre-doctoral (BES-2016-078388, funded by MCIN/AEI/10.13039/501100011033 and "ESF investing in your future"), a post-doctoral fellowship (CIAPOS/2021/117, funded by VALi+d programs from GVA and "ESF investing in your future"), and a post-doctoral fellowship from Wenner-Gren Foundations, respectively. The funders had no role in study design, data collection and analysis, decision to publish, or preparation of the manuscript.

**Competing interests:** The authors have declared that no competing interests exist.

replicative helicase and the ribonucleotide reductase complexes. We propose that these two essential complexes play non-replicative roles in preventing genetic instability in response to DNA damage by genetically regulated physical interactions.

## Introduction

Complete and accurate genome duplication relies on an exquisite spatio-temporal control of DNA synthesis. An essential and conserved factor during this process is the ring-shaped helicase MCM, which unwinds the DNA molecule to provide the template for deoxynucleoside triphosphates (dNTPs) incorporation during DNA synthesis. To ensure that replication occurs just once per cell cycle, MCM is loaded at replication origins in late mitosis and G1 as part of an inactive pre-replicative complex (preRC) and activated during S phase by the coordinated phosphorylation activities of the cyclin-dependent kinase (CDK) and the Dbf4-dependent kinase (DDK) [1–3]. In addition, CDK prevents re-replication through several overlapping mechanisms including destruction or inhibition of pre-RC components and nuclear exclusion of non-loaded MCM complexes during S and G2/M [4–8].

Another essential factor for DNA replication is the ribonucleotide reductase (RNR) complex, which catalyzes the rate-limiting step in dNTP synthesis by reducing ribonucleotides to deoxyribonucleotides. High levels of dNTPs are also induced in response to DNA damage or replication stress, likely to facilitate repair by promoting the activity of translesion synthesis (TLS) polymerases. In eukaryotic cells RNR is formed by two nonidentical dimeric subunits, which contain the catalytic and allosteric binding sites that control substrate preference and total activity (R1; $\alpha_2$), and a stable diferric tyrosyl radical cofactor essential for RNR activity (R2; $\beta_2$ or $\beta\beta'$) [9–11]. Although the minimal quaternary state of an active RNR is a tetramer ($\alpha_2\beta_2$ or $\alpha_2\ \beta\beta'$), the binding of the allosteric effectors ATP and dATP leads to the formation of different $\alpha$ oligomeric states that can modulate the reductase activity [12]. The budding yeast contains an RNR complex with some unique features. First, the R2 subunit is formed by an heterodimer consisting of Rnr2 and Rnr4, having Rnr4 an structural role [13–15]. Second, in addition to the essential R1 subunit Rnr1, yeast cells express a second non-essential R1 subunit–Rnr3 –in response to DNA damage and non-fermentable carbon sources [16,17].

Allosteric and oligomeric mechanisms are not the only ones that control RNR activity. Eukaryotic cells regulate *RNR* transcription along the cell cycle and in response to DNA damage [9]. In addition, yeast cells employ small proteins to control the activity and location of RNR components. For instance, in *Sacharomyces cerevisiae* Sml1 binds to and inhibits R1 [18,19], whereas Dif1 binds to and promotes the nuclear import of R2. Since R1 is constitutively cytoplasmic, R2 nuclear shuttling by Dif1 inhibits dNTPs synthesis [20,21]. Many of these regulatory mechanisms are up-regulated by the checkpoint kinases ATR/ATM-Chk1 in mammals and Mec1-Rad53 in yeast. Specifically, Mec1-Rad53 activates the kinase Dun1 by phosphorylation; then, Dun1 phosphorylates several substrates to promote either their inhibition (the *RNR2/3/4* transcriptional repressor Crt1) or their degradation (Sml1 and Dif1) [20,22,23]. This pathway ensures a rapid increase in the dNTP pools in response to DNA damage and replication stress, but also upon entry into S phase, when it is activated by the low dNTP levels [24]. On the contrary, mammalian cells hardly increase the pool of dNTPs in response to DNA damage, recruiting instead the RNR complex at sites of DNA damage to supply dNTPs during DNA repair [25].

Despite multiple mechanisms operating to get an accurate duplication of the genome, different DNA lesions accumulate during DNA replication that need to be repaired as faithfully

as possible to avoid the loss of genetic information. A major mechanism dealing with DNA lesions during replication is homologous recombination (HR), which in contrast to TLS is mostly error-free. An important feature of HR is that occurs at DNA repair centers that can be detected as foci by labelling with fluorescent markers DNA repair and checkpoint factors [26–28]. Interestingly, some HR proteins like Rad52 plays non-recombinogenic roles in TLS that can be visualized by an accumulation of Rad52 foci in TLS mutants, genetically colocalizing both repair processes at the DNA repair centers [29]. Recent microscopy analyses of DNA repair centers suggest the existence of a liquid droplet around the damaged DNA where most Rad52 diffuses freely by the entire compartment. In contrast, the ssDNA binding complex RPA follows the diffusion pattern of the DNA lesion, suggesting that it is mostly bound to DNA [30,31].

We have recently shown that the MCM complex plays a non-canonical function in response to replicative stress that is mediated by physical interactions with the HR proteins Rad51 and Rad52. These interactions are established in G1 at a nuclease-insoluble nucleoprotein scaffold outside of the replication origins and maintained in the S phase by DDK to assist stressed replication forks through non-recombinogenic functions [32]. To get a deeper insight into the role of MCM in response to replicative DNA damage, we have searched for MCM interactors in the absence and presence of the alkylating agent methyl methanesulfonate (MMS). Our results show that a small subpopulation of the MCM and RNR complexes interact with each other in a DNA damage and Dun1-regulated manner, migrating from the nucleus in G1 to the cytoplasm in S phase. These physical interactions 1) facilitate the release of Rad52 – but not RPA–from DNA repair centers once the lesions are repaired and 2) prevent hypermutagenesis and hyperrecombination. Importantly, these functions are not associated with the canonical ones of MCM and RNR in DNA replication and dNTP synthesis. These results uncover novel roles for MCM in the maintenance of genome integrity through physical interactions with RNR components.

## Results

### DNA damage and Dun1 regulate physical interactions between the helicase MCM and the RNR complex

To obtain a deeper insight into the roles of the MCM complex in DNA damage repair, we searched for interacting partners of Mcm4 under unperturbed conditions and MMS-induced replicative stress. For this, Mcm4-GFP was immunoprecipitated with GFP-trap beads under native conditions from asynchronous cultures treated or not with 0.025% MMS for 2 hours. Next, Mcm4-GFP interacting partners were identified by mass spectrometry-based proteomic analysis. Statistical analysis revealed known and novel Mcm4 interactors (Fig 1A). As expected, Mcm4 interacted with multiple components of the replisome regardless of the presence of DNA damage. In response to MMS, as previously shown for hydroxyurea (HU), Mcm4 interacted with Rad53 (Fig 1A) [33]. In addition, we observed MMS-specific interactions of Mcm4 with Dun1 and, with low statistical significance but high fold-increase relative to the untagged control, Rnr3. Indeed, the novel Mcm4 interactors included several factors involved in the expression and regulation of the RNR complex (the Ccr4/Not complex and Crt10) [34,35].

We decided to analyze some of these interactions by co-immunoprecipitation (CoIP) and western blot analyses. In addition to Dun1 and Rnr3, we focused on Ccr4, since it is required for MMS and HU resistance (S1 Fig) [34]. Confirming the mass spectrometry data, the Mcm4/Ccr4 interaction was detected under unperturbed and replicative stress conditions, whereas the Mcm4/Dun1 interaction was weakly detected under unperturbed conditions and induced by DNA damage (Fig 1B and 1C). Rnr3 expression is induced in response to DNA damage

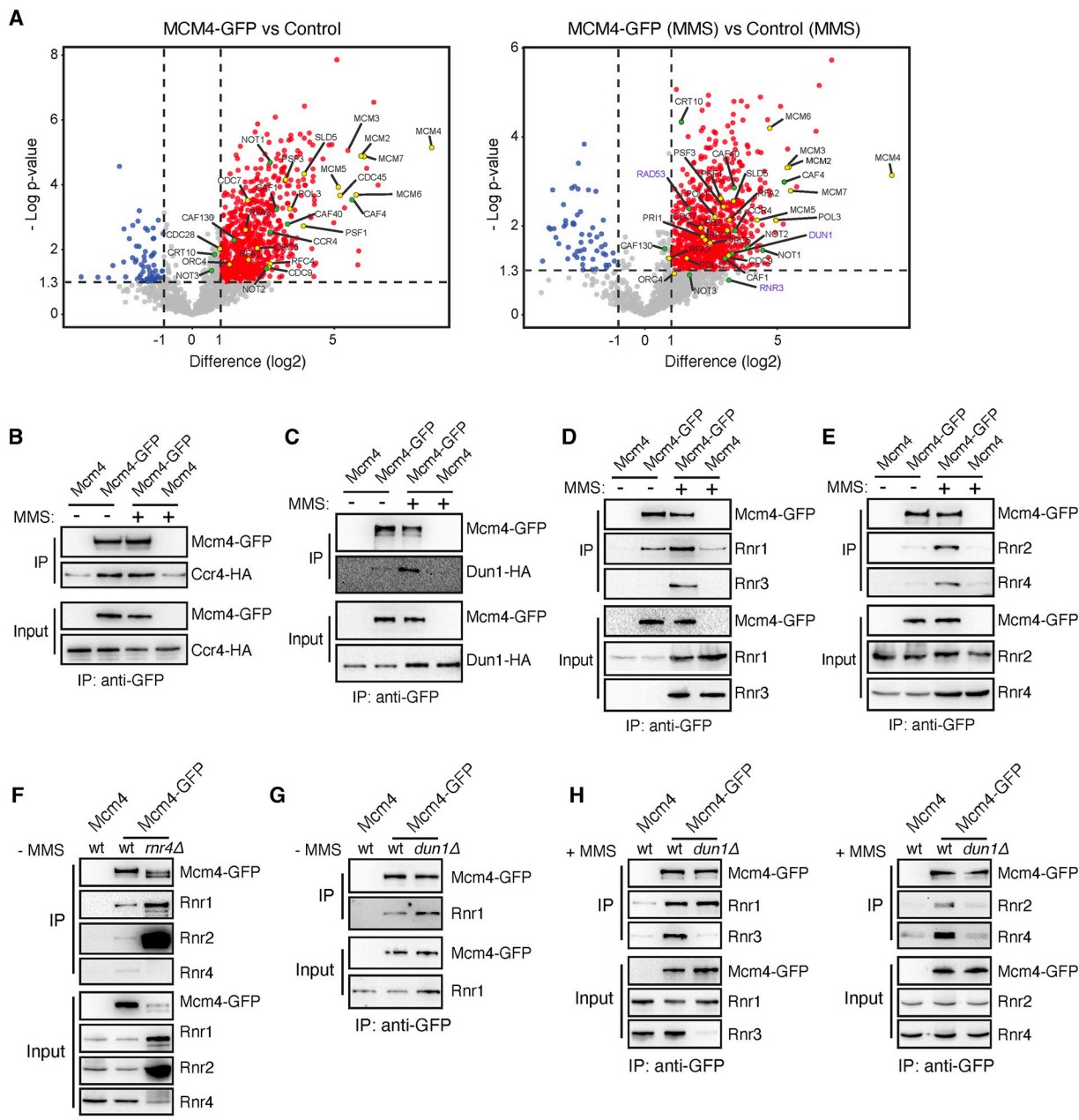

**Fig 1. DNA damage and Dun1 regulate physical interactions between the helicase MCM and the RNR complex. (A)** Volcano plots depicting statistical differences between Mcm4-GFP and non-tagged Control co-immunoprecipitated proteins in the absence (left) and presence of 0.025% MMS (right) after mass spectrometry-based proteomics analysis. Proteins with an enrichment in the Mcm4-GFP samples higher than 1 (log2) and a p-value lower than 0.05 (*t*-test) are considered positive hits (red). Each dot represents a protein. Replisome and RNR network factors are labelled in yellow and green, respectively. Data were obtained from four independent biological repeats. **(B-E)** Analyses by CoIP and western blot of the interaction of Mcm4 with Ccr4 (B), Dun1 (C), Rnr1, Rnr3 (D), Rnr2 and Rnr4 (E) in the presence and absence of DNA damage. **(F)** Rnr4 is not required for the interaction of Mcm4 with Rnr1 and Rnr2, as determined by CoIP and western blot in *rnr4Δ* and wild-type cells. **(G-H)** Dun1 is required for Rnr2 and Rnr4, but not Rnr1 to interact with Mcm4 in the absence and presence of MMS, as determined by CoIP and western blot in *dun1Δ* and wild-type cells. (B-H) Co-IPs were performed in asynchronous cultures treated or not with 0.025% MMS for 2 hours. All CoIP analyses were performed at least twice with similar results.

[36], and accordingly, the Mcm4/Rnr3 interaction was detected only upon MMS treatment (Fig 1D). These results prompted us to test by CoIP if Mcm4 interacted with the other subunits of the RNR complex, which might be missing from the proteomic analysis by technical reasons. As previously shown for Dun1, Mcm4 interacted weakly with Rnr1, Rnr2 and Rnr4 under unperturbed conditions and these interactions were increased by MMS (Fig 1D and 1E). This increase was also observed in response to HU (S2A Fig), discarding the possibility that the MCM/RNR interactions were induced by the proteotoxic effect of MMS. It is worth noting that the DNA damage-induced expression might explain totally or partially the increase in the interaction with MCM of Rnr1 and Rnr4, respectively, but not of Rnr2 (Figs 1D, 1E, and S2A). Furthermore, the MCM/Rnr1 interaction was also observed with an antibody against Mcm7 (S2B Fig), suggesting that it involves the MCM complex. Finally, we tested the interaction between MCM and RNR in the absence of Rnr4, the only non-essential subunit of the constitutive RNR complex. The amount of Rnr1 and, above all, Rnr2, increased in the *rnr4Δ* background, and proportionally the interaction with Mcm4 (Fig 1F).

Dun1 stimulates the synthesis of dNTPs by the RNR complex during DNA replication and DNA damage [37–39]. Thus, we tested if Dun1 was required for the MMS-induced MCM/RNR interactions. The absence of Dun1 did not affect the Mcm4/Rnr1 interaction, whereas it was required for Mcm4 to interact with Rnr2 and Rnr4 (Rnr3 is not expressed in the absence of Dun1) (Fig 1G and 1H). Altogether, these results uncover a novel set of physical interactions between the MCM and RNR complexes that are modulated by DNA damage and Dun1.

## The MCM/RNR interactions encompass a small and differentially regulated subpopulation of the MCM and RNR complexes

To obtain a deeper insight into the regulation of the MCM/RNR interaction, we explored the effect of perturbing the location of the MCM and R2 pools with two different strategies. The accumulation of the R2 complex in the nucleus requires the importing activity of Dif1 and the nuclear anchoring activity of Wtm1 [20,40]; accordingly, R2 accumulates in the cytoplasm during the whole cell cycle in a *dif1Δ wtm1Δ* double mutant, as can be determined by counting the percentage of cells that display a predominant accumulation of a Rnr4-Cherry chimera in the nucleus along the cell cycle in the absence and presence of MMS (Fig 2A and 2B, top panel) [20]. Additional expression of Mcm4-GFP allowed us to follow the localization of the MCM complex in *dif1Δ wtm1Δ* cells, which did not change as compared to wild-type cells (Fig 2A and 2B, bottom panel). Likewise, we forced the accumulation of the MCM complex in the nucleus by tagging the Mcm4-GFP chimera with a strong nuclear localization signal (Fig 2A and 2C, top panel; Mcm4-GFP-NLS). As control, we used a strain expressing a mutant nuclear localization signal (Mcm4-GFP-nls3A2). This tagged Mcm4 protein exited from the nucleus with a slight delay as compared to the wild-type Mcm4 protein (Fig 2A and 2C, top panel). Constitutive accumulation of the MCM complex in the nucleus in *MCM4-GFP-NLS* cells did not affect the dynamics of Rnr4-Cherry along the cell cycle (Fig 2A and 2C, bottom panel). These results suggest that the interaction between MCM and R2 encompasses a small subpopulation of each complex.

We took advantage of this strategy to address how the location of MCM and RNR affected the MCM/RNR interactions. We observed that retaining MCM into the nucleus in the Mcm4-GFP-NLS strain did not affect the interactions of MCM with all RNR components both in the presence and absence of MMS (Fig 2D), despite Rnr1 and Rnr3 localize at the cytoplasm [21]. Actually, MCM/RNR interactions were also detected in G1-arrested cells that physiologically accumulate MCM in the nucleus (Fig 2E). Furthermore, the interactions of MCM with Rnr2 and Rnr4 were not altered when the Rnr2/Rnr4 heterodimer was forced to remain at the

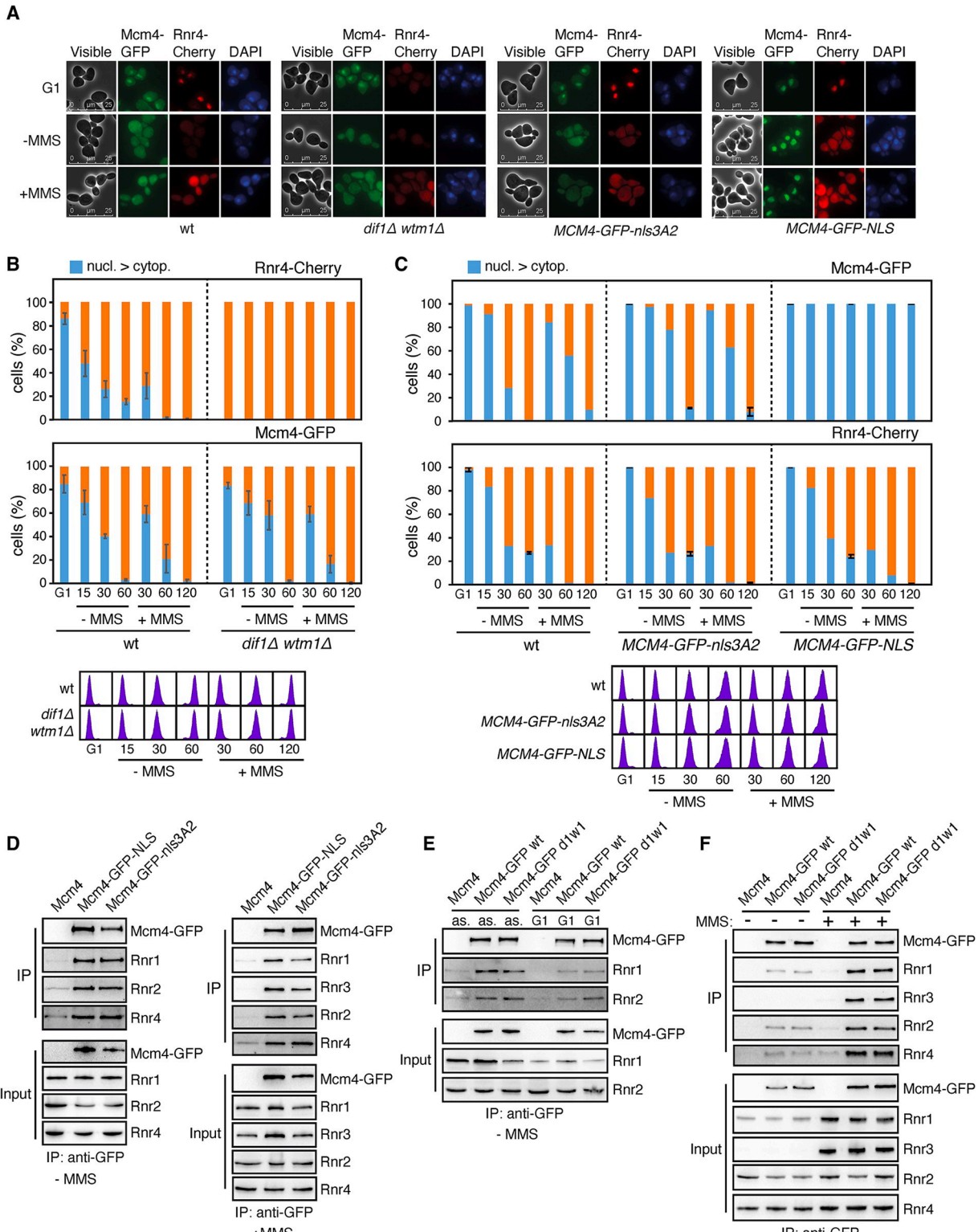

**Fig 2. The MCM/RNR interaction is independent of the subcellular location of MCM and RNR. (A-C)** The interaction between MCM and Rnr2/Rnr4 encompasses a small fraction of each complex. Mcm4-GFP and Rnr4-Cherry subcellular distribution in *dif1Δ wtm1Δ* (A-B) and *Mcm4-GFP-NLS* (A, C) cells synchronized in G1 and released into fresh medium in the absence and presence of 0.025% MMS. *Mcm4-GFP-nls3A2* and wild-type cells were included as control. Cell cycle progression was followed by cell sorting analysis. The number of cells with a signal intensity in the nucleus higher than in the cytoplasm was counted directly on the processed samples. The mean and SEM of three (error

bars) or two independent experiments are shown, with a total number of approximately 100 cells analyzed at each time and experiment. **(D-F)** The subcellular location of the MCM and RNR complexes does not affect the MCM/RNR interactions. Analyses by CoIP and western blot of the interaction of Mcm4 with RNR components in *Mcm4-GFP-NLS*, *Mcm4-GFP-nls3A2* (D) and *dif1Δ wtm1Δ* (E-F) cells. Co-IPs were performed in asynchronous (D-F) or G1-arrested (E) cultures treated or not with 0.025% MMS for 2 hours. All CoIP analyses were performed at least twice with similar results. Note that CoIPs with and without MMS in (D) were independent experiments and cannot be compared with each other.

cytoplasm in the *dif1Δ wtm1Δ* double mutant (Fig 2F), even in G1-arrested *dif1Δ wtm1Δ* cells in which MCM and Rnr2/Rnr4 accumulate at different cellular compartments (Fig 2E). Therefore, the integrity of the MCM/RNR interactions is independent of the subcellular location of the major MCM and RNR pools. Altogether, these results suggest that the MCM/RNR interactions encompass a small and differentially regulated subpopulation of the MCM and RNR complexes.

## Mcm4-VC/Rnr4-VN interactions accumulate in the nucleus in G1 and migrate to the cytoplasm during S/G2 independently of the MCM and Rnr2/Rnr4 complexes

To study the MCM/RNR interactions in live cells, we used bimolecular fluorescence complementation (BiFC). In this assay, potential interacting partners are endogenously tagged with the N-terminal (VN) and C-terminal (VC) domain of the fluorescent protein Venus. Physical interaction between the partners can reconstitute a fluorescent signal that allows to detect the cellular location and quantify the strength of the interaction [41]. Since we could only construct Mcm4-VC, we tagged Dun1, Ccr4 and the subunits of the RNR complex with the VN domain. Only Rnr4-VN generated a fluorescent signal specific of the presence of Mcm4-VC (S3A Fig).

We used the BiFC assay to study the location of the Mcm4/Rnr4 interaction during the cell cycle. The Venus signal was followed by fluorescence microscopy in cells synchronized in G1 and released into fresh medium. Cells were grouped according to the bud-to-mother size ratio to compare similar stages of DNA replication with and without DNA damage (S3B Fig). Since the signal was spread homogeneously both in the nucleus and the cytoplasm (S3A Fig), the concentration of the Mcm4-VC/Rnr4-VN complex in the nucleus relative to the cytoplasm along the cell cycle was calculated as the ratio between the Venus signal in an area of the nucleus and the Venus signal in an equivalent area of the cytoplasm. The fluorescent signal accumulated in the nucleus in G1, and changed gradually to the cytoplasm as cells progressed into S and G2/M phases under unperturbed conditions (Fig 3A, left panels). This behavior was similar to that previously reported for both the MCM complex and the R2 subunit [6,7,20,21], although with a different kinetics: it migrated faster than Rnr4 and slower than Mcm4 (Fig 3A, compare left with middle and right panels, respectively).

When cells were released from G1 into S phase in the presence of MMS, the accumulation of the R2 subunit in the cytoplasm is accelerated to augment the pool of dNTPs required for efficient DNA repair (Fig 3A, middle panels) [21]. In the case of MCM, the kinetics from the nucleus to the cytoplasm was delayed in the presence of MMS (Fig 3A, right panels), which is consistent with the maintenance of MCM at chromatin when cells replicate in the presence of MMS [32]. On the contrary, the kinetics of nucleus-to-cytoplasm transfer of the Mcm4-VC/Rnr4-VN complex was unaffected by the presence of MMS (Fig 3A, left panels). Therefore, the Mcm4-VC/Rnr4-VN complex and the individual MCM and R2 complexes display different kinetics of mobilization from the nucleus to the cytoplasm, which may reflect different regulation or susceptibility to the same regulatory mechanism.

The Mcm4-VC/Rnr4-VN fluorescence signal was apparently similar in the absence and presence of MMS, in contrast to our biochemical analyses with Mcm4-GFP and Rnr4 (Fig 1E).

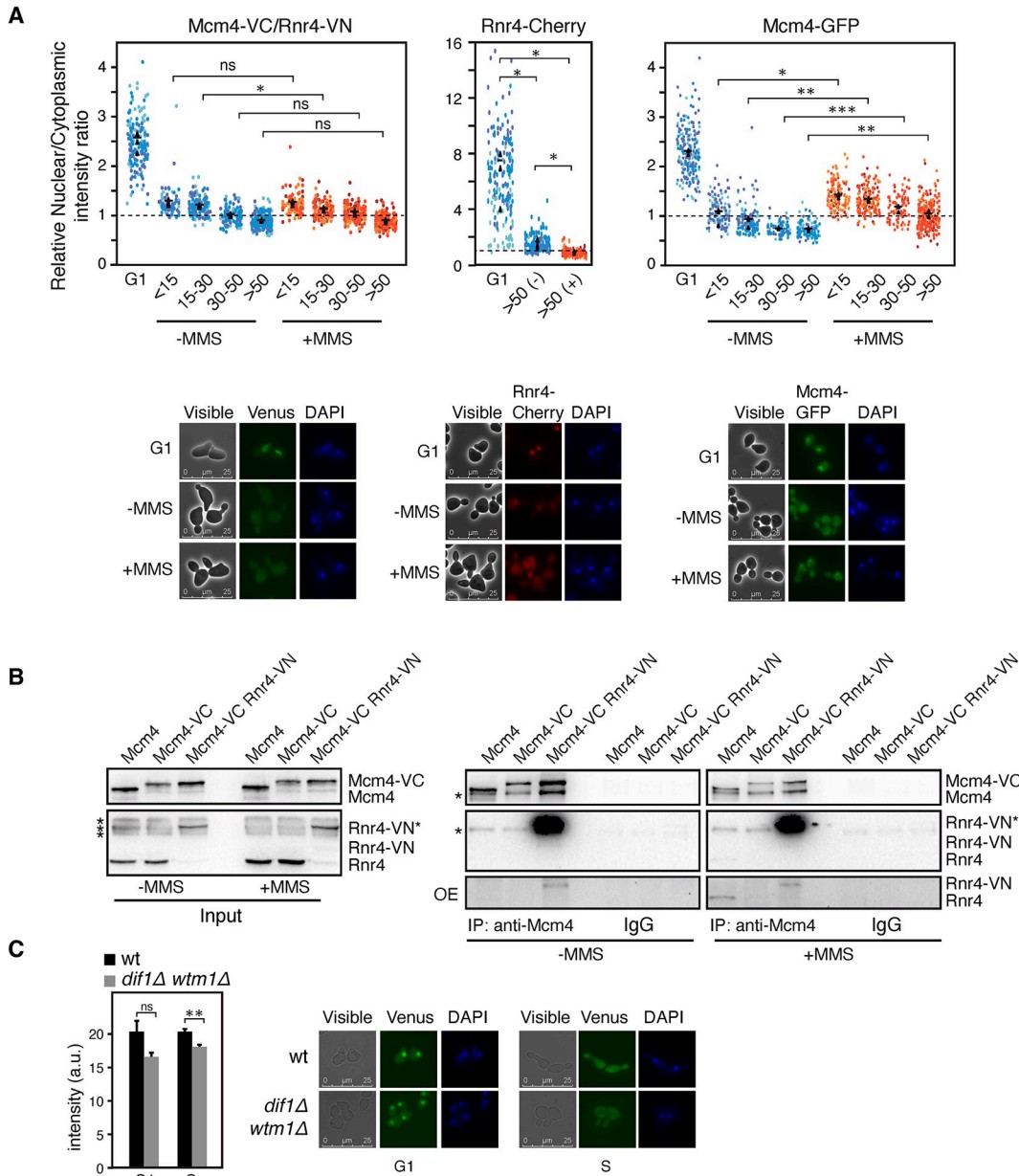

**Fig 3. Characterization of the intensity and location of the Mcm4-VC/Rnr4-VN interaction. (A)** The Mcm4-VC/Rnr4-VN interaction accumulates in the nucleus in G1 and migrates to the cytoplasm in S/G2 independently of the MCM and Rnr2/Rnr4 complexes. BiFC analysis of the location of the Mcm4-VC/Rnr4-VN interaction during the cell cycle in cells synchronized in G1 and released into fresh medium in the absence and presence of 0.025% MMS (left panel). Cells were grouped according to the bud-to-mother size ratio to compare similar stages of DNA replication with and without DNA damage. The concentration of Mcm4/Rnr4 complex in the nucleus relative to the cytoplasm along the cell cycle was calculated as the ratio between the Venus signal in an area of the nucleus and the Venus signal in an equivalent area of the cytoplasm. The subcellular location of the Rnr2/Rnr4 (middle panel) and MCM (right panel) complexes in cells synchronized in G1 and released into fresh medium in the absence and presence of 0.025% MMS was followed by the signal of Rnr4-Cherry and Mcm4-GFP, respectively. SuperPlots show the mean and SEM from three independent experiment (represented by color dots). Representative images of cells in G1 and S phase (with and without MMS) are shown for each strain. **(B)** Physical interactions between Mcm4-VC and Rnr4-VN, as determined by CoIP and western blot in asynchronous cultures treated or nor with 0.025% MMS for 2 hours. Wild-type and Mcm4-VC-expressing cells were included as controls. Asterisks indicate unspecific bands. Rnr4-VN* indicates a slow-migrating Rnr4-VN form. CoIP analyses were performed at twice with similar results. IP samples were run in different gels due to space limitations but treated in parallel, and therefore the signals are comparable. An over-exposition (OE) of the Rnr4/Rnr4-VN IP signals is shown in the bottom panels. **(C)** The Mcm4-VC/Rnr4-VN interaction is independent of the subcellular location of the MCM and RNR complexes. The Venus signal (plot)

and the subcellular location (images) of the Mcm4-VC/Rnr4-VN interaction were determined in *dif1Δ wtm1Δ* and wild-type cells synchronized in G1 and released into fresh medium for 60 minutes (S phase). The mean and SEM of three independent experiments are shown. (A, C) Asterisks indicate statistically significant differences according to an unpaired two-tailed Student's *t*-test (one, two and three asterisks represent *P*-values <0.05, <0.01 and <0.001, respectively).

This was confirmed by quantifying the Mcm4-VC/Rnr4-VN fluorescence signal by cell sorting analysis in cells synchronized in G1 and released into fresh medium in the absence and presence of MMS. Actually, the Venus signal hardly changed along the cell cycle, despite the amount of Rnr4 augmented in response to MMS during S phase (S3C Fig). These results suggest that the Mcm4/Rnr4 interaction is either stabilized or blind to MMS in the presence of Mcm4-VC and/or Rnr4-VN. To distinguish between these possibilities, we compared by CoIP and western blot the Mcm4/Rnr4 and Mcm4-VC/Rnr4-VN interactions with and without MMS using antibodies against Mcm4 and Rnr4 that allow to detect untagged and tagged proteins. This analysis showed two findings: First, most Rnr4-VN migrates with an apparently higher molecular weight (Rnr4-VN*) in the PAGE gel (Fig 3B, input); second, this potentially post-translationally modified Rnr4-VN* chimera has a MMS-independent strong affinity for Mcm4-VC (Fig 3B, IP panels). However, unmodified Rnr4-VN also interacted with Mcm4-VC in a MMS-independent manner despite was hardly detected in the input (Fig 3B, over-exposed Rnr4/Rnr-VN signals in the IP panels), suggesting that the strong interaction with Mcm4 is associated with the Rnr4-VN chimera that for unknown reasons becomes modified.

These results prompted us to ask if the Mcm4-VC/Rnr4-VN signal was a reliable readout of the Mcm4/Rnr4 interaction. Since our previous analysis showed that the Mcm4/Rnr4 interaction is independent of the cellular location of the major MCM and RNR pools (Fig 2D–2F), we determined the intensity and location of the Mcm4-VC/Rnr4-VN signal in wild-type and *dif1Δ wtm1Δ* cells synchronized in G1 and released into S phase. The signal was hardly affected in *dif1Δ wtm1Δ* cells (Fig 3C), despite most of the MCM and R2 complexes become spatially separated in G1 in this mutant. Interestingly, the location of the Venus signal was located at the nucleus in both strains (Fig 3C), suggesting that it is determined by the MCM location. Therefore, although artefactually augmented, the Mcm4-VC/Rnr4-VN signal provides an *in vivo* evidence for the nucleus-to-cytoplasm migration of the MCM/R2 complex. In addition, it further supports an independent regulation of the MCM/RNR interactions from the individual MCM and RNR pools.

## The MCM/RNR interactions are partially disrupted by the Mcm4-VC chimera

The analysis of the Mcm4-VC/Rnr4-VN interaction by CoIP and western blot revealed a partial loss of interaction in cells expressing only Mcm4-VC (Figs 3B and S2B). We extended this biochemical analysis to the rest of RNR subunits in the presence of MMS (Fig 4). This analysis showed that Mcm4-VC was partially defective in interacting with all RNR components (Fig 4). In addition, it revealed that the Rnr4-VN forms bind strongly not only to Mcm4-VC but also to Mcm4, and stimulates to a similar extent the interaction of Rnr2 with Mcm4 and Mcm4-VC. Thus, the massive recruitment of Rnr4-VN by Mcm4-VC can be partially but not completely explained by a stabilization of the interaction through the VC and VN domains. Rnr4-VN also reduced the interaction of Mcm4 with Rnr1 and Rnr3, may be due to the binding of multiple Rnr2/Rnr4-VN heterodimers. These results open the possibility of using the *MCM4-VC* strain to study the effect of partially disrupting the MCM/R2 interactions. Importantly, to confirm that a defect induced by Mcm4-VC is due to a loss of interactions with RNR

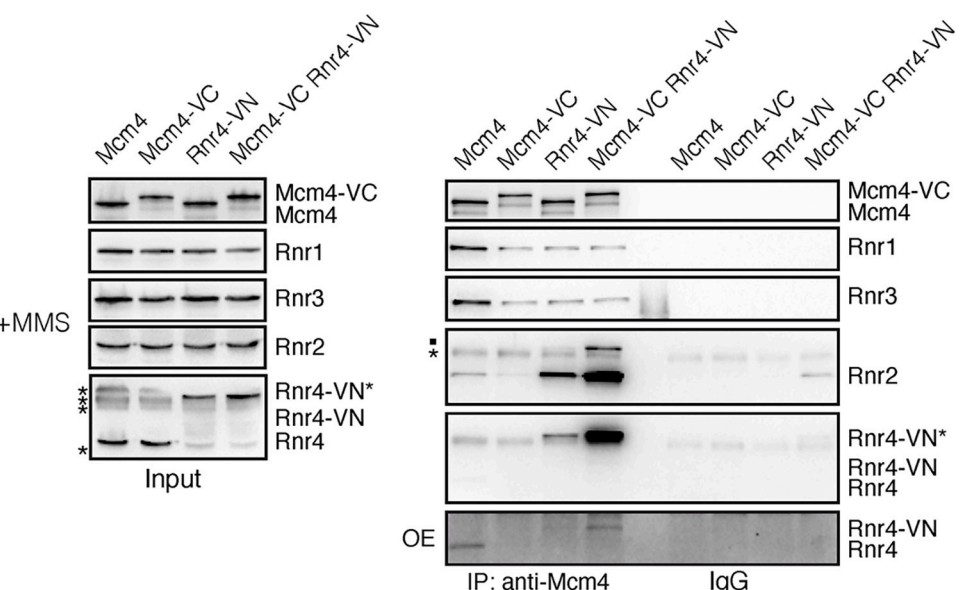

**Fig 4. The MCM/RNR interactions are partially disrupted by the Mcm4-VC chimera.** Effect of tagging Mcm4 and Rnr4 with VC and VN, respectively, on the physical interactions between the MCM and RNR complexes, as determined by CoIP and western blot in asynchronous cultures treated with 0.025% MMS for 2 hours. Asterisks indicate unspecific bands. A dot indicates cross-hybridization of the anti-Rnr2 antibody with Rnr4. Rnr4-VN* indicates a slow-migrating Rnr4-VN form. An over-exposition (OE) of the Rnr4/Rnr4-VN IP signals is shown in the bottom panel. CoIP analyses were performed twice with similar results.

components and not to another MCM function, this defect should be at least partially recovered in cells expressing Mcm4-VC and Rnr4-VN where the MCM/R2 interactions are "artefactually" restored. This recovery experiment is essential to genetically associate a phenotype with the loss of MCM/RNR interactions, as the tagging of the Mcm4 with VC can alter some other functions hampering the interpretation of the results. In addition, it is important to demonstrate that the *MCM4-VC* allele is recessive for the analyzed phenotype to rule out the possibility that this phenotype was due to a dominant negative effect of Mcm4-VC separable of its interaction with RNR and titrated away by Rnr4-VN in the double mutant *MCM4-VC RNR4-VN*.

A limitation of this approach is that the Rnr4-VN chimera seems not to be completely functional as inferred from the HU sensitivity displayed by the *RNR4-VN* mutant (S4A Fig), even though the drop in the levels of dNTPs was subtle (S4B Fig). To test if this slight decrease in the pool of dNTPs had additional consequences for the cell, we measured the percentage of petite colonies, which are formed by cells that do not contain functional mitochondria and cannot grow on a non-fermentable carbon source. Mitochondria are particularly sensitive to alterations in the pool of dNTPs, as the one induced by Sml1 elimination that causes a 2-3-fold decrease in petite formation (S4C Fig) [18]. We observed that the expression of Rnr4-VN duplicated the frequency of petites and that this increase was suppressed by eliminating Sml1 (S4C Fig), consistent with a loss of functionality of the Rnr4-VN subunit. Therefore, unless otherwise specified, we performed the rest of analyses in cells transformed with a centromeric plasmid expressing *RNR4* from its own promoter (p314RNR4), which rescued the loss of dNTPs, the mitochondrial dysfunction and the HU sensitivity of the *RNR4-VN* mutant (S4B, S4D and S4E Fig). It is worth noting that the frequency of petite formation in *RNR4-VN* cells transformed with p314RNR4 was 4 times lower than in the wild type expressing two *RNR4* alleles (S4D Fig). This suggests that the RNR complex is more active in the former, although

this difference was not technically detected by measuring the levels of dNTPs (S4B Fig). Importantly, expression of an additional *RNR4* allele did not alter the pattern of MCM/RNR interactions (S4F Fig), which was similarly affected by Mcm4-VC and Rnr4-VN in the absence and presence of DNA damage (S4G Fig). It is also worth noting that a putative suppression of Mcm4-VC-associated defects by co-expressing Rnr4-VN might be partially compromised by the loss of stoichiometry of the interactions in the *MCM4-VC RNR4-VN* mutant (the Mcm4-VC/Rnr4-VN interaction keeps being the major one in the *MCM4-VC RNR4-VN* mutant transformed with p314RNR4; S4F Fig).

## The MCM/R2 interactions are required for efficient Rad52, but not RPA, release from DNA repair centers after DNA damage repair

Our biochemical analyses suggest that the MCM/RNR interactions play a role in the DNA damage response, as they are induced by genotoxic agents and are dependent on Dun1 (Fig 1). To address this possibility, we analyzed the *MCM4-VC* mutant. Expression of Mcm4-VC caused HU and MMS sensitivity, partial DNA damage checkpoint activation as determined by Rad53 phosphorylation and Sml1 degradation and Rad52 foci accumulation under unperturbed conditions (S4A and S5A–S5C Figs). Actually, this analysis also showed a slight degradation of Sml1 in cells expressing Rnr4-VN (S5B Fig), consistent with their low levels of petite cells (S4D Fig). The *MCM4-VC* allele was recessive for the accumulation of spontaneous DNA damage as demonstrated by co-expressing a wild-type copy of Mcm4 under its own promoter from a centromeric plasmid (S4A and S5E Figs, raffinose); however, this phenotype was not prevented by the expression of Rnr4-VN (S5A–S5C Fig). It is likely that they result from defects in the MCM helicase activity, although these defects are insufficient to affect bulk DNA replication in accordance with the slight checkpoint activation (S5D Fig). In any case, the fact that they also arise in cells expressing Rnr4-VN indicates that are not due to a loss of MCM/RNR interactions.

Taking into account this result, we decided to analyze the effect of expressing Mcm4-VC in the repair of a DSB at the *MAT* locus, which occurs by intra-molecular HR with the DNA sequence of either *HMLα* or *HMRa* through a mechanism that does not require the MCM helicase activity (Fig 5A) [42]. For this, the expression of the endonuclease HO (under control of the *GAL1* promoter) was induced for 2 hours in galactose and then repressed by adding glucose to allow the repair of the DNA break. The accumulation and recombinational repair of the DSB was followed by the formation and resolution of Rad52-YFP foci, respectively. Whereas wild-type and *RNR4-VN* cells hardly formed detectable Rad52 foci, consistent with a quick and efficient repair, the *MCM4-VC* mutant accumulated specific HO-induced Rad52 foci (from 12% of cells before HO expression due to the spontaneous accumulation of DNA damage to 27% of cells 1 hour after HO repression) that 6 hours later had not reached the spontaneous level yet (Fig 5B). Importantly, expression of Rnr4-VN in the *MCM4-VC RNR4-VN* mutant slightly reduced the peak of HO-induced Rad52 foci and accelerated their resolution, which was complete 1 hour after the peak (note that the level of spontaneous Rad52 foci limited the efficiency of foci resolution to ~50%) (Fig 5B and 5C). Furthermore, *MCM4-VC* was recessive for the accumulation of HO-induced Rad52 foci (S5E Fig). These results suggest that the MCM/R2 interactions are required for DSB repair.

To further confirm this apparent DNA repair defect in the *MCM4-VC* mutant, we directly followed the kinetics of DSB formation and repair by physical analysis of HR intermediates under similar experimental conditions. Unexpectedly, except for a slight shift in the election of the donor sequence, the *MCM4-VC* strain displayed wild-type kinetics of DSB formation and repair (Fig 5D and 5E). Accordingly, it did not accumulate phosphorylated Rad53, not even

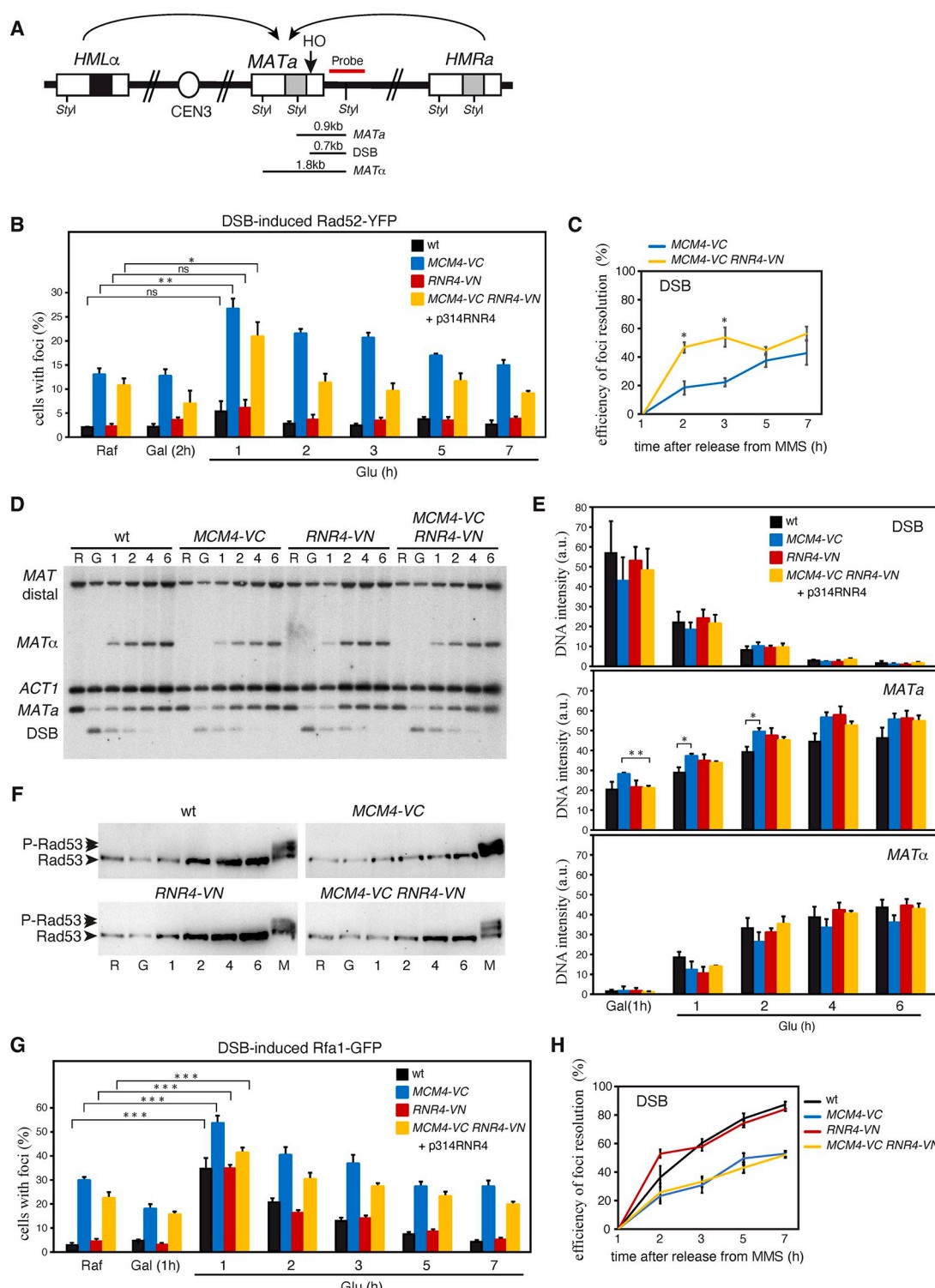

**Fig 5. The MCM/RNR interactions are required for efficient Rad52, but not RPA, release from DSB-induced DNA repair centers after DNA damage repair. (A)** Scheme of the *MAT* switching recombination system. The position of the HO and *Sty*I sites and the length of the DNA fragments generated after DSB formation and repair with the *HMRa* and *HMLα* cassettes are shown. **(B-C)** The loss of MCM/RNR interactions in *MCM4-VC* cells impairs DSB-induced Rad52 foci resolution. Raffinose growing cells were shifted to galactose-containing medium for 2 hours to induce the expression of the endonuclease HO (under control of the *GAL1* promoter), and then to glucose-containing medium to repress it and allow the repair of the DSB. The number of cells with Rad52-YFP was counted directly on the processed samples under the microscope. (B) The mean and SEM

of three independent experiments are shown, with a total number of approximately 100 cells analyzed at each time point and experiment. (C) The efficiency of foci resolution was determined as (100 –percentage of cells with foci at each time point x 100/ maximal percentage of cells with foci during the time course). **(D-F)** The MCM/RNR interactions are not required for DSB repair. Kinetics of HO-induced DSB repair at the *MAT* locus. Cells were treated as in (B), and samples were taken at the indicated time points for DNA and protein extraction. (D) DSB formation and repair by HR with the information of the *HMRa* and *HMLα* cassettes was followed by southern blot as the appearance of specific DNA fragments. (E) Quantification of DSB formation (top panel, DSB) and repair with *HMRa* (middle panel, *MATa*) or *HMLα* (middle panel, *MATα*) were calculated by normalizing the corresponding signal to *ACT1*, and this value to the ratio *MATa/ACT1* in raffinose. The mean and SEM of three independent experiments are shown. (F) DNA damage checkpoint activation was followed by western blot against Rad53. An asynchronous culture treated with 0.025% MMS for 1 hour was included as positive control (M). **(G-H)** The loss of MCM/RNR interactions does not impair DSB-induced RPA foci resolution. Cells expressing Rfa1-GFP were treated and processed as in (B-C). Asterisks indicate statistically significant differences according to an unpaired two-tailed Student's *t*-test (one, two and three asterisks represent *P*-values <0.05, <0.01 and <0.001, respectively). All experiments were performed with cells transformed with plasmid p314RNR4.

spontaneously under these experimental conditions (Fig 5F). Therefore, the MCM/RNR interactions are not required for DSB repair, but they facilitate the release of Rad52 from the DNA repair center once the DNA lesion is repaired.

Since Rad52 diffuses freely by the DNA repair center whereas RPA follows the diffusion pattern of the DNA lesion [30,31], we asked if the MCM/R2 interactions were specifically required to maintain the dynamics of the Rad52 liquid droplets, while being unnecessary to retain RPA once the repair was completed. For this, we repeated the kinetics of HO-induced DSB repair in cells expressing Rfa1 (the largest subunit of the RPA complex) fused to GFP. In this case, and consistent with previous studies showing a higher level of DNA repair foci detection with Rfa1 than with Rad52 [29,43], a peak of RPA foci 1 hour after glucose addition could be detected in wild-type and *RNR4-VN* cells (Fig 5G). As mentioned for Rad52 foci, the level of spontaneous RPA foci limited the efficiency of foci resolution to ~50% in the *MCM4-VC* strains versus ~80% in the wild type. In contrast to the wild type, the slope of foci resolution before reaching the 50% was slower in the *MCM4-VC* strains (Fig 5H). This might be due to the concomitant accumulation of spontaneous foci during the resolution of DSB-induced foci in the latter. Importantly, the kinetics of foci resolution was similar in the *MCM4-VC* and *MCM4-VC RNR4-VN* strains (Fig 5H), indicating that the MCM/RNR interactions are not involved in the release of RPA from DNA repair centers. Therefore, the MCM/R2 interactions are not required for the recombinational repair of DSBs, but for the efficient release of Rad52 from the DNA repair centers once the lesions are repaired.

### Persistence of Rad52 at DNA repair centers in cells defective in MCM/R2 interactions is associated with genetic instability

Next, we asked if the MCM/R2 interactions were also required for Rad52 release from DNA repair centers induced in response to genotoxic agents that generate replicative stress. For this, cells were synchronized in G1 and released for 1 hour in the presence of MMS or HU; then, the drug was eliminated and cells were released into fresh medium to allow DNA repair. The accumulation and repair of ssDNA was followed by the formation and resolution of Rad52-YFP foci (Fig 6A and 6C). In response to MMS, all strains displayed a peak of approximately 70% of cells with Rad52 foci one hour after MMS release that reflects the recombinational repair of the ssDNA gaps left behind the fork as a consequence of the bypass of the blocking lesions [43,44]. The kinetics of Rad52 foci resolution was slower in the *MCM4-VC* mutants as compared to the *RNR4-VN* and wild-type strains. This defect was more severe in *MCM4-VC* than in *MCM4-VC RNR4-VN*, although the difference was not statistically significant (Fig 6A and 6B, left panels). In response to HU, none of the strains formed Rad52 foci in the presence of the drug (Fig 6A, right panel), consistent with the replicative checkpoint

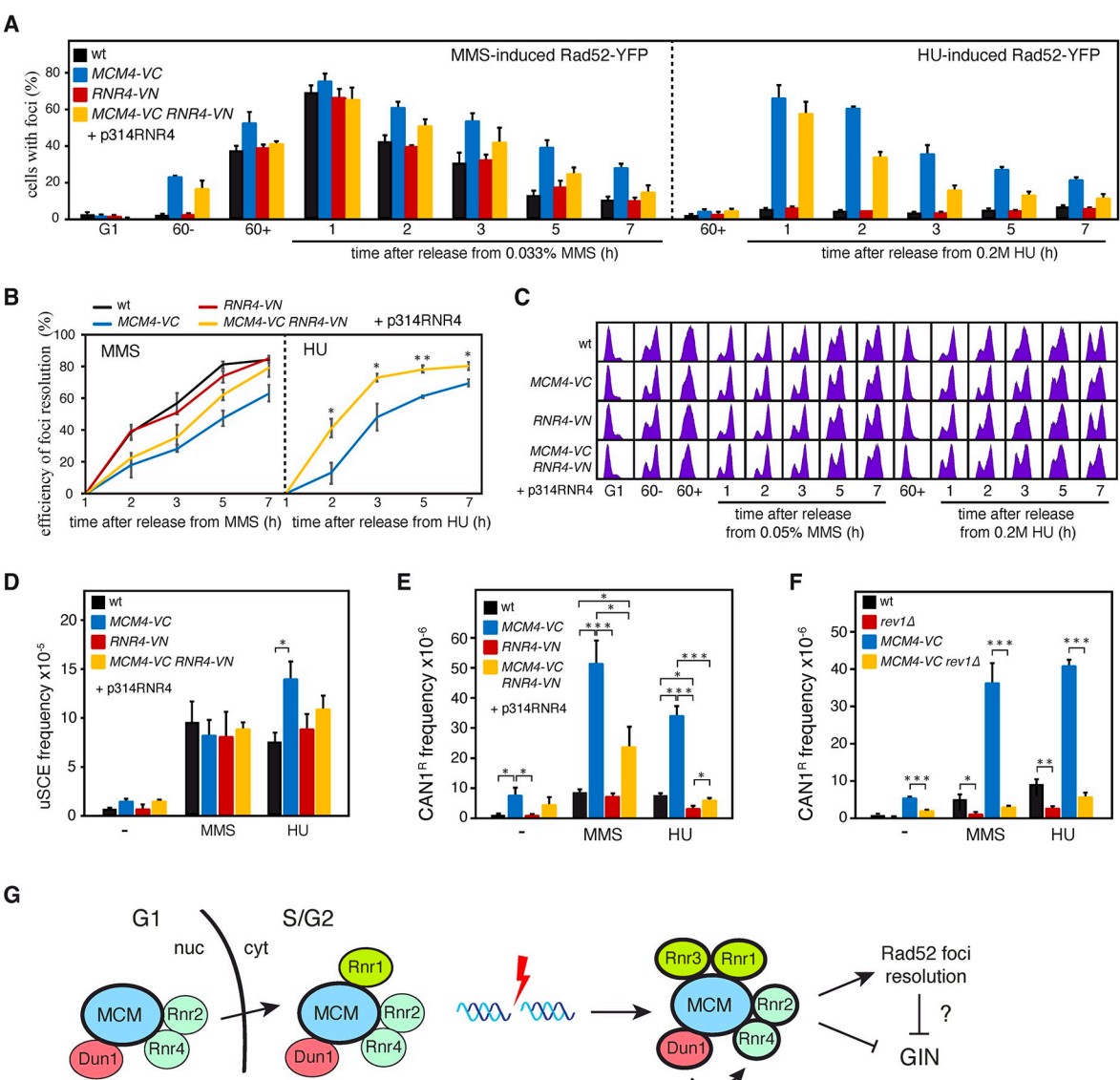

**Fig 6. Persistence of Rad52 at DNA repair centers in cells defective in MCM/R2 interactions is associated with genetic instability.**
**(A-C)** The loss of MCM/RNR interactions in *MCM4-VC* cells impairs MMS and HU-induced Rad52 foci resolution. G1-synchronized cells were released into S phase in the presence of 0.05% MMS or 0.2M HU for 1 hour, washed to remove the genotoxic agent and released into fresh medium. G1-synchronized cells were also released in the absence of DNA damage for 1 hour to determine the frequency of cells with spontaneous Rad52 foci (60-). The number of cells with Rad52-YFP was counted directly on the processed samples under the microscope. (A) The mean and SEM of three independent experiments are shown, with a total number of approximately 100 cells analyzed at each time point and experiment. (B) The efficiency of foci resolution was determined as (100 –percentage of cells with foci at each time point x 100/ maximal percentage of cells with foci during the time course). (C) Cell cycle progression was followed by cell sorting analysis. **(D)** The loss of MCM/RNR interactions in *MCM4-VC* cells does not reduce MMS- and HU-induced uSCE. The frequency of HR was determined from colonies grown without or with genotoxic agents (0.0075% MMS or 50mM HU). The mean and SEM of 4–7 fluctuations tests are shown. **(E-F)** The loss of MCM/RNR interactions in *MCM4-VC* cells increases spontaneous, MMS- and HU-induced Rev1-dependent mutagenesis. The frequency of mutagenesis was determined from colonies grown without or with genotoxic agents (0.0075% MMS or 50mM HU). The mean and SEM of 4–7 fluctuations tests are shown. (B, D-F) Asterisks indicate statistically significant differences according to a unpaired two-tailed Student's *t*-test (one, two and three asterisks represent *P*-values <0.05, <0.01 and <0.001, respectively). (A-E) The experiments were performed with cells transformed with plasmid p314RNR4. **(G)** Model: Under unperturbed conditions, MCM interacts with Rnr1 and weakly with Rnr2/Rnr4 and Dun1. The MCM/Rnr2/Rnr4 interactions accumulate in the nucleus in G1 and are mobilized to the cytoplasm during S phase, where Rnr1 resides and it is likely interacting with MCM (whether Dun1 interacts with MCM in the nucleus and/or the cytoplasm is unknown). These physical interactions encompass small subpopulations of MCM and RNR, are independent of the major subcellular locations of these two complexes, augment in response to DNA damage and depend on Dun1 in the case of Rnr2/Rnr4. Through mechanisms that are not associated with alterations in the levels of dNTPs, the MCM/ RNR interactions facilitate the release of Rad52 from the DNA repair center once the lesions are repaired and prevent genetic instability (GIN) by hypermutagenesis and HR.

preventing the formation of Rad52 foci in response to DSBs and replicative damage during S phase [43,45]. Indeed, Mcm4-VC expressing cells were completely functional in checkpoint activation (S6A Fig). However, and in contrast to the wild-type strain, the *MCM4-VC* mutants accumulated a high percentage of cells with Rad52 foci once they reached G2/M (Fig 6A, right panel), consistent with the HU triggering the breakage of replication forks by the tagged MCM helicase. Importantly, the resolution of these Rad52 foci was faster in the *MCM4-VC RNR4-VN* mutant than in the *MCM4-VC* mutant (Fig 6B, right panel).

To rule out that this Rad52 foci resolution defect was due to impaired ssDNA gap filling, we analyzed the frequency of MMS and HU-induced unequal sister chromatid exchange (uSCE). The *MCM4-VC* strain was proficient in MMS and HU-induced HR, with wild-type levels in response to MMS and a two-fold increase in response to HU that was not observed in the *MCM4-VC RNR4-VN* mutant (Fig 6D). In addition, we tested if Rad52 foci persistence in the *MCM4-VC* mutant could be due to impaired TLS [29]. For this, we measured the frequency of forward mutagenesis at the *CAN1* locus (selected as canavanine-resistant cells). The *MCM4-VC* was proficient in TLS; actually, it displayed a strong increase in spontaneous (10-fold), MMS (6-fold) and HU-induced (5-fold) Rev1-dependent mutagenesis (Fig 6E and 6F), suggesting that replicative DNA lesions are bypassed by Pol ζ-mediated TLS. Importantly, the *MCM4-VC* mutation was recessive for the increase in mutagenesis (S6B Fig) and this increase was partially (MMS) or totally (HU) suppressed in the double mutant *MCM4-VC RNR4-VN* (Fig 6E). Therefore, Rad52 foci persistence by disruption of the MCM/R2 interactions is not due to the lack of repair of the ssDNA fragments generated by replicative stress. This further supports the idea that the MCM/R2 interactions facilitate the release of Rad52 from the DNA repair centers once the lesions are repaired, and importantly, show that the unscheduled persistence of Rad52 at these centers is associated with genetic instability.

The network of DNA damage and Dun1-regulated interactions of MCM with RNR and several of its regulators opens the possibility that the MCM complex operates as a hub regulating the RNR activity. We thus asked if the genetic instability phenotypes associated with the loss of MCM/RNR interactions might be due to alterations in the pool of dNTPs. To test this, we measured the levels of dNTPs in the *MCM4-VC* strains. These levels augmented approximately two times in the *MCM4-VC* mutant in the absence of genotoxic agents (S7A Fig). This increase is likely influenced by the accumulation of spontaneous DNA damage in this mutant as supported by the reduction in the frequency of petite formation and the accumulation of Rnr3 (S4D and S7B Figs); however, the pool of dNTPs was slightly but significantly lower in the double *MCM4-VC RNR4-VN* than in the single *MCM4-VC* mutant, pointing to a putative role for the MCM/R2 interactions negatively regulating the levels of dNTPs. In contrast to the situation in the absence of genotoxic agents, the partial loss of MCM/R2 interactions in the *MCM4-VC* mutant did not affect the concentration of dNTPs in the presence of 0.05% MMS (S7A Fig), discarding the possibility that Rad52 foci persistence and genetic instability in this mutant were associated with unbalanced dNTPs levels.

## Discussion

The essential function of the MCM and RNR complexes is related to their activities providing the ssDNA templates and dNTPs for DNA replication, respectively [1–3,9–11]. The synthesis of dNTPs by RNR is also critical for an efficient DNA damage response, whereas the role of MCM for DNA repair is limited to some processes like break-induced replication and replication fork assistance under stress conditions [32,46]. Here, we provide evidence for the MCM complex acting as a hub for DNA damage-regulated physical interactions with the RNR complex and some of its regulators, including Rad53 and Dun1. These MCM/RNR interactions

encompass a small and differentially regulated subpopulation of MCM and RNR complexes. Using a *MCM4-VC* allele that partially disrupts the MCM/RNR interactions, we genetically demonstrate that these interactions facilitate the release of Rad52 from DNA repair centers once the lesions are repaired and prevent hypermutagenesis through mechanisms that are not associated with alterations in the levels of dNTPs (Fig 6G).

By mass spectrometry and western blot analyses from Mcm4-GFP immunoprecipitated samples we have uncovered a network of physical interactions of the MCM complex with the RNR complex and several of its regulators. They include Ccr4/Not, Crt10, and importantly, the checkpoint kinases Rad53 and Dun1, which are master regulators of RNR controlling its expression, location and activity [20,22,23,47]. Except for the Ccr4/Not complex, the rest of interactors augmented their interactions with MCM in response to DNA damage and, in the case of R2, this increase depended on Dun1. Therefore, MCM seems to act as a hub to recruit (directly or mediated by bridging proteins) RNR components and regulators that is reinforced by DNA damage. We cannot, though, discard the possibility that the interactions take place between some subcomplexes or non-canonical forms of the MCM and RNR complexes. Importantly, these MCM/RNR interactions encompass a small and differentially regulated fraction of MCM and RNR complexes, as evidenced by the fact that their integrity and location is not affected by forcing the location of the major pools of MCM and RNR at different subcellular compartments. A simple explanation for the functionality of these interactions would be that they modulated the RNR activity in response to DNA damage, and actually the *MCM4-VC* mutant displays a slight increase in the pool of dNTPs that seems to be associated with a partial loss of the MCM/R2 interactions. However, this increase was not observed in response to 0.05% MMS, which causes more DNA damage and checkpoint activation than the expression of the Mcm4-VC protein. Therefore, at this point, we cannot conclude that the MCM/RNR interactions have a role in controlling the levels of dNTPs.

DNA repair occurs at membrane-less compartments that accumulate DNA repair and checkpoint proteins [28]. Because these centers are formed in response to DNA damage and are disassembled once the lesions are repaired, they have been routinely used to measure the kinetics and efficiency of DNA repair. Unexpectedly, we show that the recombination protein Rad52 persists at the DNA repair centers in cells displaying a partial disruption of the MCM/R2 interactions, despite the lesions that triggered the formation of the foci were efficiently repaired. This behaviour is not general, as the ssDNA binding factor RPA was efficiently released under similar experimental conditions. A major difference between Rad52 and RPA in yeast is that the motion of Rad52 at the foci follows a liquid-phase model where the protein explores the entire droplet, whereas RPA follows the diffusion pattern of the damaged chromatin consistent with most RPA bound to ssDNA [30,31]. Remarkably, Rad52 molecules are retained at the foci by an attractive potential, suggesting that DNA is dispensable to maintain the phase separation once a critical concentration is reached [31]. Supporting this model, Rad52 is retained at the foci in the *MCM4-VC* mutant despite the DNA lesions are repaired and there is not ssDNA to bind.

These results suggest that the MCM/R2 interactions might be required to maintain the attractive potential that retains Rad52 at the droplet. At this point we do not have any clue about how this is achieved, except that is not mediated by alterations in the pool of dNTPs. Some examples have been reported in which RNR components impact cellular functions independently of dNTPs, like a non-yet defined mitochondrial function of Rnr3 in yeast or a nuclear non-proliferative activity of an hexameric state of RNR-α interacting to and inhibiting the translocase ZRANB3 in mammal [17,48]. The MCM/R2 interactions are unlikely regulating Rad52 persistence through direct interactions, because the Rad52 foci are nuclear structures that appear only during the S/G2/M phases [26] in which the MCM/R2 interactions

accumulate in the cytoplasm. Actually, this cytoplasmic location rules out the possibility that the MCM/RNR interactions were related to the ones occurring at the nuclease-insoluble nucleoprotein scaffold where MCM interacts with Rad51 and Rad52 [32]. Alternatively, the MCM/RNR interactions might trigger a signaling cascade to regulate Rad52 dynamics at foci, or might affect Rad52 persistence more indirectly. MCM is required for the maintenance of stress granules, which also contain RNR components [49]. MCM/RNR interactions in these or similar membrane-less organelles might regulate the availability of mRNAs and/or proteins required for controlling the integrity of DNA repair centers. Further studies will be required to unveil the regulation and mechanisms of action of the MCM/RNR interactions in genome integrity.

Rad52 persistence at foci in the *MCM4-VC* mutant is associated with genetic instability, despite HR and TLS are proficient. First, it causes Rev1-mediated hypermutagenesis. Apart from its major role in HR, Rad52 cooperates with the TLS machinery through non-recombinogenic functions, including the recruitment to damaged chromatin of the Rad6/Rad18 complex to promote PCNA ubiquitylation and further binding of TLS polymerases [29]. In this context, excessive Rad52 might lead to an unscheduled accumulation of TLS polymerases. However, the frequency of HR should be reduced if the filling of the ssDNA fragments is channeled to TLS. Alternatively, excessive Rad52 at the DNA repair centers might promote Rev1-mediated mutagenic HR. In vitro D-loop extension by the replicative polymerase Pol δ is highly stimulated by the replication processivity factor PCNA. However, TLS polymerases extend the D-loop more efficiently than Pol δ in the absence of the processivity factor, a situation stimulated by Rad51 inhibition of PCNA loading [50]. Since Rad52 promotes the formation of the Rad51/ssDNA nucleofilament [51,52], excessive Rad52 at the DNA repair centers might hamper PCNA loading and facilitate mutagenic recombination. In apparent contrast with this possibility, MMS-induced HR in the *MCM4-VC* mutant is independent of Rev1 (S8 Fig). However, it is likely that in this background Pol δ gets back its function. Unfortunately, we cannot determine the dependency on HR of the hypermutagenic phenotype of the *MCM4-VC* mutant because the absence of HR causes a strong increase in Rev1-mediated mutagenesis [29,53,54]. In addition to hypermutagenesis, the choice of the template used to repair the lesion by HR seems to be slightly affected (Fig 5E), which might explain the HU-induced increase in uSCE in the *MCM4-VC* mutant (Fig 6D). Altogether, these results suggest that excessive Rad52 at the DNA repair centers alters the output of the repair processes leading to genetic instability, may be by altering the correct binding of DNA repair proteins at DNA lesions. Alternatively, Rad52 accumulation at DNA repair centers might affect its function at other compartments, like its interactions with MCM at the nucleoprotein scaffold. However, we cannot exclude the possibility that the defect in Rad52 release from the repair centers and the increase in genetic instability were independent of each other.

In summary, our work reveals a network of physical interactions between specifically regulated subpopulations of the MCM and RNR complexes with non-canonical roles in the maintenance of genome integrity. Considering the conservation of these complexes, it will be worth to address in future studies if these interactions are conserved and what impact they have in the genome integrity of higher eukaryotes.

## Source data

Raw data are shown in S9 Fig and S3 Table. The mass spectrometry data have been deposited to the ProteomeXchange Consortium [55] via the PRIDE partner repository with the data set identifier PXD045914.

## Materials and methods

### Yeast strains, plasmids and growth conditions

All *Saccharomyces cerevisiae* strains used are haploid derived from W303. Yeast strains used in this study are listed in S1 Table. Most strains were generated by genetic crosses. Tagged and deletion strains were constructed by a PCR-based strategy [56]. The original uSCE strains [57,58] were backcrossed at least four times into the W303 background.

pWJ1344 is a centromeric plasmid expressing *RAD52-YFP* [59]. pGAL-HO is a multi-copy plasmid expressing the endonuclease HO from the *GAL1* promoter [60]. p314RNR4 and p316MCM4 are centromeric plasmids expressing *RNR4* and *MCM4* from their own promoters, respectively (this study). pFA6A-3HA- HIS3MX6 [56], pKT209 [61], pFA6Am-Cherry-NatMX6 [62], pFA6A-VC- HIS3, pFA6A-VN- KanMX6 [41], pFA6AKanMX6-NLSsv40 and pFA6AKanMX6-svnls32 (this study) are plasmids for protein tagging with HA, eGFP, mCherry, the C- and N-terminal halves of Venus, the nuclear localization signal of SV40 (NLSsv40) and a non-functional NLSsv40 [6], respectively. pFA6AKanMX4 [63], pAG32 [64] are plasmids for gene disruption with kanamycin and hygromycin, respectively. p314RNR4 and p316MCM4 were constructed by inserting a 1.8 kb (*RNR4*) or a 3.6 kb (*MCM4*) PCR fragment at the *Cla*I-*Bam*HI site or the *Hind*III-*Xho*I site of pRS314 and pRS316, respectively [65]. pFA6AKanMX6-NLSsv40 and pFA6AKanMX6-svnls32 were constructed by inserting the *nls* sequences generated as gBlocks DNA fragments (IDT) (see S2 Table for sequences) at the *Nde*I-*Bam*HI site of pFA6AKanMX6. Oligonucleotide sequences are shown in S2 Table.

Cells were grown at 30°C in supplemented minimal medium (SMM) with glucose, galactose or raffinose as carbon source, except in Figs 2A–2C, 3A, 3C and S3 where cells were grown in rich medium (YPAD). The mediums were supplemented with adenine (x3) to reduce cell autofluorescence in fluorescence microscopy experiments. For G1 synchronization, cells were grown to mid-log phase and α-factor was added twice at 60 min intervals at either 2 (*BAR1* strains) or 0.25 µg/ml (*bar1Δ* strains). Then, cells were washed three times and released into fresh medium with 50 µg/ml pronase in the absence or presence of HU or MMS at the indicated concentrations. The drug was eliminated by washing cells three times before being released into fresh medium, and in the case of MMS, cells were treated with 2.5% sodium thiosulfate for 2–3 minutes to inactivate it before washing.

### Petite formation analysis

Strains were first streaked on YPGlycerol (3%) plates to start with wild-type cells and then on YPAD plates to allow the accumulation of petite cells. The frequency of petites was determined by fluctuation tests from six independent colonies of similar size. They were plated with the appropriate dilutions onto YPAD (total cells) and then replicated to YPGlycerol. The frequency of petites was calculated using the median of the six colonies. To have a more accurate value, the mean and SEM of 3 independent fluctuation tests are given.

### Cell cycle analysis

Cell cycle was followed by DNA content analysis, which was performed by flow cytometry as reported previously [66]. Cells were fixed with 70% ethanol, washed with phosphate-buffered saline (PBS), incubated with 1 mg of RNaseA/ ml PBS, and stained with 5 µg/ml propidium iodide. Samples were sonicated to separate single cells and analyzed in a FACSCalibur flow cytometer.

## DNA damage sensitivity

MMS and HU sensitivities were determined by spotting ten-fold serial dilutions of the same number of mid-log growing cells onto medium with or without the drug.

## Genetic recombination and mutagenesis analyses

HR was determined by measuring the frequency of His+ recombinants generated by uSCE in chromosomal-integrated systems [57], whereas mutagenesis was determined by measuring the frequency of forward mutagenesis at the *CAN1* locus (selected as canavanine-resistant cells). Recombination and mutagenesis frequencies were determined by fluctuation tests as previously reported [29]. Cells from six independent colonies of similar size and isolated on medium with either +/- MMS/HU were plated with the appropriate dilutions onto SMM without histidine, SMM without arginine but containing 60 μg/ml canavanine and SMM to calculate recombinants, mutants and total viable cells (as colony-forming units), respectively. For each test, the frequency of HR and mutation was calculated using the median of recombinants/mutants and the mean of total cells. To have a more accurate value, the mean and SEM of at least 3 independent fluctuation tests are given.

## DSB repair analysis

The kinetics of HO-induced DSB repair at the *MAT* locus was followed by southern blot. Briefly, total DNA from 10 ml samples was extracted using a zymolyase-SDS standard protocol, digested with *Sty*I and run into 0.8% TAE 1x agarose gels. Gels were blotted onto Hybond-XL membranes and hybridized with specific $^{32}$P-labeled probes for the *MAT* locus and the *ACT1* gene amplified by PCR from genomic DNA (see S2 Table for oligonucleotides). Signals were acquired in a Fuji FLA5100 and quantified with the ImageGauge analysis program.

## DNA repair foci analysis

The percentage of cells with RPA, Rad52 and Rad54 foci was determined as described previously [29]. Cells expressing Rfa1-YFP or transformed with plasmid pWJ1344 (expressing Rad52-YFP) were grown in liquid culture under the indicated conditions, fixed with 2.5% formaldehyde in 0.1M potassium phosphate pH 6.4 for 10 minutes, washed twice with 0.1M potassium phosphate pH 6.6 and resuspended in 0.1M potassium phosphate pH 7.4. Finally, cells were fixed with 80% ethanol for 10 minutes, resuspended in H$_2$O and visualized with a Leica CTR6000 fluorescence microscope. The percentage of cells with foci was counted directly on the processed samples under the microscope. A total number of approximately 100 cells were analyzed for each time point and experiment.

## Mcm4-GFP, Rnr4-Cherry and the Mcm4-VC/Rnr4-VN analysis

The subcellular location and quantification of Mcm4-GFP, Rnr4-Cherry and the Mcm4-VC/Rnr4-VN interaction was determined by fluorescence microscopy. DAPI-stained cells were fixed as indicated for the DNA repair foci analysis and visualized with a Leica CTR6000 fluorescence microscope. The florescent signal at the selected areas was quantified on acquired images using the ImageJ software. The total fluorescent signal of cells expressing Mcm4-GFP, Rnr4-Cherry and Mcm4-VC plus Rnr4-VN was quantified by flow cytometry. For this, cells were fixed with 2.5% formaldehyde in PBS for 10 minutes, washed (twice) and resuspended in PBS, and analyzed in a FACSCalibur flow cytometer.

## dNTPs analysis

Cells from mid-log phase cultures (O.D. ~0.5) treated or not with 0.05% MMS were harvested on nitrocellulose filters (Millipore, AAWP02500), resuspended immediately in an ice-cold lysis solution (12%TCA, 15mM MgCl$_2$), mix in a vortex for 15 min at 4°C and then centrifuged at 14,000 rpm for 1 min at 4°C. The supernatant was neutralized with a dichloromethane-trioctylamine mix and analyzed as described previously [67].

## Coimmunoprecipitation analysis

CoIP was performed with 100 ml samples from mid-log phase cultures (O.D. ~0.45) lysed with a Multi Beads Shocker (Yasui Kikai) at 4°C in 0.5 ml NP40 lysis buffer (50mM Tris pH 7.5, 150mM NaCl, 1% NP40) with protease inhibitors (1mM PMSF, 2mM DTT and Roche Complete EDTA free) and 1 volume of glass beads. Lysates were cleared by two consecutive centrifugation steps for 5 min at 1000g at 4°C and mixed with MNaseI (15mM Tris, 50mM NaCl, 1.4mM CaCl$_2$, 0.2mM EDTA, 0.2mM EGTA pH 8.0) or benzonase buffer (50mM Tris pH 8.0, 1mM MgCl$_2$) and either 2.5 u MNaseI or 25 u benzonase and incubated for 20 min at 37°C. Samples were centrifuged for 15 min at 13000 g at 4°C and the total amount of protein at the supernatant was quantified by a Bradford assay. Similar amount of proteins (~12–16 mg) were used for CoIPs. For GFP-based CoIPs, samples were incubated overnight at 4°C with 20 µl of GFP trap magnetic beads (Chromotek) in NP40 lysis buffer, and washed extensively (5 times) with NP40 lysis buffer. For Mcm4 and Mcm7 CoIPs, samples were first pre-cleared during 2h at 4°C with 20 µl protein G dynabeads (10004D, Invitrogen). Then, samples were incubated first with 3 µg/ml of antibody at 4°C overnight, then with 20 µl protein G dynabeads 2h at 4°C, and finally washed extensively with NP40 lysis buffer. Immunoprecipitated samples were eluted with Laemmli buffer and analyzed by western blot with the corresponding antibodies.

## Mass spectrometry

Mcm4-GFP immunoprecipitated samples (see coimmunoprecipitation analysis) were additionally washed five times with 50mM Tris pH 7.5, 150mM NaCl and three times with ammonium bicarbonate 50 mM, and then treated with 250ng trypsine (Promega) overnight at 30°C. The supernatant was separated from the beads by centrifugation through 0.45µM filters (Millipore, UFC30HV00), and transferred to Epperdorf LoBind 1.5 mL microcentrifuge tubes. Then, samples were acidified with 2% Trifluoroacetic acid (Sigma) and desalted with C-18 stage tips as previously described [68].

## Mass spectrometry data acquisition

The resulting peptides were analysed by nanoscale LC-MS/MS using an EASY-nLC 1000 system (Proxeon, Odense, Denmark) connected to a Q-Exactive Orbitrap (Thermo Fisher Scientific, Germany) through a nano-electrospray ion source. The Q-Exactive was coupled to a 15 cm analytical column with an inner-diameter of 75 µm, in-house packed with 1.9 µm C18-AQ beads (Reprospher-DE, Pur, Dr. Manish, Ammerbuch-Entringen, Germany). The liquid chromatography gradients was from 2% to 30% acetonitrile in 0.1% formic acid with a length of 95 min followed by column re-equilibration at a flow rate of 200 nL/min. The mass spectrometer was operated in a Data-Dependent Acquisition (DDA) mode with a Top-7 method and a scan range of 400–2000 m/z. Full-scan MS spectra were acquired at a target value of $3 \times 10^6$ and a resolution of 70,000, and the Higher-Collisional Dissociation (HCD) tandem mass spectra (MS/MS) were recorded at a target value of $1 \times 10^5$ and with a resolution of 35,000, and isolation window of 2.2 m/z, and a normalized collision energy (NCE) of 25%. The minimum AGC

target was $1\times10^3$. The maximum MS1 and MS2 injection times were 20 and 120 ms, respectively. The precursor ion masses of scanned ions were dynamically excluded (DE) from MS/MS analysis for 60 s. Ions with charge 1, and > 6, were excluded from triggering MS2 analysis.

## Mass spectromertry data analysis

Mass spectrometry Raw data were analysed using MaxQuant Software version 1.5.3.30 using default settings from [69] with the following modifications. We performed the search against an *in silico* digested UniProt reference proteome for *Saccharomyces cerevisiae S288c (24th March 2017)* including canonical and isoform sequences. Digestion with Trypsin/P was used allowing 4 missed cleavages. Oxidation (M), Acetyl (Protein N-term) and Phospho (STY) were allowed as variable modifications with a maximum number of 5. Carbamidomethyl (C) was disabled as a fixed modification. Label-Free Quantification was enabled, not allowing Fast LFQ. Match between runs was performed with 0.7 min match time window and 20 min alignment time window. All peptides were used for protein quantification. All tables were written.

Maxquant out was subsequently analyzed with the Perseus computational platform (v 1.5.5.3) [70]. LFQ intensity values were log2 transformed and potential contaminants and proteins identified by site only or reverse peptide were removed. Samples were grouped in experimental categories and proteins not identified in 4 out of 4 replicates in at least one group were also removed. Missing values were imputed using normally distributed values with a 1.8 downshift (log2) and a randomized 0.3 width (log2) considering whole matrix values. Statistical comparisons between groups were performed using two-tailed t-tests.

## Western blot

Protein samples were resolved by 10% (CoIP analyses) or 8% (Rad53 phosphorylation and Rnr3 quantification) SDS-PAGE, probed with antibodies against GFP (632381, Clontech), Mcm4 (sc-166036, Santa Cruz), Mcm7 (sc-6688, Santa Cruz), Rnr1, Rnr2, Rnr3, Rnr4 (Nguyen 1999), Pgk1 (22C5D8, Invitrogen), Rad53 (Abcam, ab104232) or Sml1 [47], and detected with fluorophore conjugate secondary (Rnr3 quantification) or peroxidase-conjugate antibodies (rest). Yeast protein extracts to analyse Rad53 phosphorylation and Rnr3 levels were prepared using the TCA protocol as described [71]. The immunoluminescent signal was generated with either the WesternBright ECL (Advansta) or the Clarity Western ECL Substrate (BioRad) kit, acquired in a ChemiDoc MP image system and quantified with the Image Lab software (Biorad). The fluorescent signal was visualized and quantified using the Odyssey infrared Imaging System (Licor).

## Supporting information

**S1 Fig. Ccr4 and Rnr4 help to tolerate MMS and HU sensitivity.** The absence of Ccr4 and Rnr4 causes MMS and HU sensitivity as determined by ten-fold serial dilutions at the indicated concentrations. The experiment was repeated twice with similar results.
(TIFF)

**S2 Fig. Characterization of the MCM/RNR interactions. (A)** HU promotes MCM/RNR interactions, as determined by CoIP and western blot in asynchronous cultures treated with 0.2M HU for 2 hours. Untreated and treated cells with 0.025% MMS for 2 hours were included as control. **(B)** Mcm7 interacts with Rnr1, as determined by CoIP and western blot in asynchronous cultures of wild-type cells. This interaction is lost in the *MCM-VC* mutant. Asterisks represent unspecific bands.
(TIFF)

**S3 Fig. Subcellular location and intensity of the Mcm4-VC/Rnr4-VN interaction. (A-C)**
BiFC analysis of the subcellular location and intensity of the Mcm4VC/Rnr4-VN interaction
during the cell cycle in cells synchronized in G1 and released into fresh medium in the absence
and presence of 0.025% MMS. (A) Representative images of dividing cells expressing
Mcm4-VC and/or Rnr4-VN. (B) Intensity of the Mcm4VC/Rnr4-VN fluorescence signal in
the nucleus and cytoplasm of cells grouped according to the bud-to-mother size ratio to com-
pare similar stages of DNA replication with and without DNA damage. The Venus signal was
calculated in equivalent areas of the nucleus and the cytoplasm. The SuperPlot shows the mean
and SEM from three independent experiment (represented by color dots). (C) Intensity of the
Mcm4-VC/Rnr4-VN fluorescence signal as determined by cell sorting analysis. Cells were
grouped in G1, early and late S phase (eS and lS) and G2/M according to the presence and size
of the bud. The amount of the MCM and Rnr2/Rnr4 complexes was followed by quantifying
the GFP signal in cells expressing Mcm4-GFP and Rnr4-GFP under the same experimental
conditions. The mean and SEM of 4–7 fluctuations tests are shown.
(TIFF)

**S4 Fig. Supplementary information related to Fig 4. (A)** The expression of Mcm4-VC or
Rnr4-VN causes HU and MMS sensitivity as determined by ten-fold serial dilutions at the
indicated concentrations (top panels). The *MCM4-VC* allele is recessive for HU and MMS sen-
sitivity, as determined in cells transformed with a centromeric plasmid expressing Mcm4 from
its own promoter (p316MCM4) or an empty vector (pRS316) (bottom panels). The experi-
ments were repeated twice with similar results. **(B)** Levels of dNTPs in *RNR4-VN* and wild-
type cells transformed or not with plasmid p314RNR4. The mean and SEM of four indepen-
dent experiments are shown. **(C-D)** Frequency of petite formation in the indicated strains.
The mean and SEM of 3 fluctuations tests are shown. **(E)** The HU sensitivity of the *RNR4-VN*
mutant can be complemented with a Rnr4 expressing plasmid as determined by ten-fold serial
dilutions at the indicated concentrations. The experiments were repeated twice with similar
results. **(F-G)** Analysis of the MCM/RNR interactions in cells expressing Mcm4-VC and/or
Rnr4-VN transformed with plasmid p314RNR4. Effect of tagging Mcm4 and Rnr4 with VC
and VN, respectively, on the physical interactions between the MCM and RNR complexes in
cells transformed with plasmid p314RNR4, as determined by CoIP and western blot in asyn-
chronous cultures treated (F) or not (G) with 0.025% MMS for 2 hours. Asterisks indicate
unspecific bands. Rnr4-VN* indicates a slow-migrating Rnr4-VN form. An over-exposition
(OE) of the Rnr4 IP signals is shown in the bottom panel. CoIP analyses were performed twice
with similar results.
(TIFF)

**S5 Fig. The Mcm4-VC chimera leads to DNA damage accumulation. (A-B)** Rad53 phos-
phorylation (A) and Sml1 levels (B) at asynchronous cultures of the indicated strains trans-
formed with plasmid p314RNR4 in the absence of genotoxic agents. A wild-type culture
incubated with 0.025% MMS for 1 hour was included as positive control. **(C)** Spontaneous
accumulation of Rad52 foci at asynchronous cultures of the indicated strains transformed with
plasmid p314RNR4. **(D)** Cell cycle progression of *MCM-VC* and wild-type strains synchro-
nized in G1 and released into S phase, as determined by cell sorting analysis. (E) The
*MCM4-VC* allele is recessive for the accumulation of spontaneous and HO-induced Rad52
foci, as determined in cells transformed with a centromeric plasmid expressing Mcm4 from its
own promoter (p316MCM4) or an empty vector (pRS316). The kinetics of HO-induced
Rad52 foci formation and resolution was performed as indicated in Fig 5B. (C, E) The mean
and SEM of three independent experiments are shown. Asterisks indicate statistically signifi-
cant differences according to an unpaired two-tailed Student's *t*-test (three asterisks represent

a *P*-values <0.001).
(TIFF)

**S6 Fig. Characterization of checkpoint and mutagenesis in *MCM4-VC* cells. (A)** Checkpoint activation is proficient in *MCM4-VC* mutants. Checkpoint activation in *MCM4-VC* and *RNR4-VN* mutants, as determined by western blot against Rad53 in asynchronous cultures treated with 0.2M HU for 60 minutes. Cell cycle progression was determined by cell sorting analysis. The P-Rad53/Rad53 ratio was determined by dividing the upper bands signal (Rad53-P) by the lower band signal (Rad53). Experiments were performed with cells transformed with plasmid p314RNR4. **(B)** The *MCM4-VC* allele is recessive for hypermutagenesis, as determined in *MCM4-VC* and wild-type cells transformed with a centromeric plasmid expressing Mcm4 from its own promoter (p316MCM4) or an empty vector (pRS316). The frequency of mutagenesis was determined from colonies grown with 0.0075% MMS or 50mM HU. The mean and SEM of three independent experiments are shown. Asterisks indicate statistically significant differences according to an unpaired two-tailed Student's *t*-test (three asterisks represent a *P*-values <0.001).
(TIFF)

**S7 Fig. dNTP levels in *MCM4-VC* and *RNR4-VN* mutants. (A)** Levels of dNTPs in *MCM4-VC* and *RNR4-VN* mutants in the absence and presence of 0.05% MMS, as determined in mid-log phase asynchronous cultures of the indicated strains transformed with plasmid p314RNR4. The mean and SEM of four independent experiments are shown. Asterisks (relative to the wild type) and dots (between the indicated values) show statistically significant differences according to an unpaired two-tailed Student's *t*-test (one, two and three asterisks represent *P*-values <0.05, <0.01 and <0.001, respectively). **(B)** The amount of Rnr3 is increased in *MCM4-VC* cells, as determined by western blot analysis from asynchronous cultures of *MCM4-VC* and wild-type cells. The amount of Rnr3 was normalized to the amount of Pgk1. The mean and SEM of three independent experiments are shown. Asterisks indicate statistically significant differences according to an unpaired two-tailed Student's *t*-test (two asterisks represent a *P*-values <0.01).
(TIFF)

**S8 Fig. MMS-induced uSCE in the *MCM4-VC* mutant is independent of Rev1.** The frequency of HR was determined from colonies grown in the presence of 0.0075% MMS. The mean and SEM of 4 fluctuations tests are shown.
(TIFF)

**S9 Fig. Raw data for Figure panels.** Original blots for the indicated figure panels are shown.
(PDF)

**S1 Table. Saccharomyces cerevisiae strains used in this study.** Strains, genotypes, references and Figures panels where they have been used are indicated.
(DOCX)

**S2 Table. Oligonucleotydes used in this study.** DNA sequences and Figures panels where they have been used are indicated.
(DOCX)

**S3 Table. Raw data for Figure plots.** Raw values to build graphs in the indicated Figure panels are shown.
(XLSX)

## Acknowledgments

We thank JoAnne Sttube and Michael Lisby for various strains and reagents, Cristina González Garrido for the construction of p316MCM4 and Andrei Chabes and Andrés Clemente Blanco for critical reading of the manuscript.

## Author Contributions

**Conceptualization:** Félix Prado.

**Data curation:** Aurora Yáñez-Vilches, Antonia M. Romero, Marta Barrientos-Moreno, Román González-Prieto, Sushma Sharma.

**Formal analysis:** Aurora Yáñez-Vilches, Antonia M. Romero, Marta Barrientos-Moreno, Román González-Prieto, Sushma Sharma.

**Funding acquisition:** Alfred C. O. Vertegaal, Félix Prado.

**Investigation:** Aurora Yáñez-Vilches, Antonia M. Romero, Marta Barrientos-Moreno, Esther Cruz, Román González-Prieto, Sushma Sharma.

**Project administration:** Alfred C. O. Vertegaal, Félix Prado.

**Resources:** Aurora Yáñez-Vilches, Antonia M. Romero, Marta Barrientos-Moreno.

**Supervision:** Félix Prado.

**Validation:** Aurora Yáñez-Vilches, Antonia M. Romero, Marta Barrientos-Moreno, Esther Cruz, Román González-Prieto, Sushma Sharma, Alfred C. O. Vertegaal, Félix Prado.

**Visualization:** Aurora Yáñez-Vilches, Félix Prado.

**Writing – original draft:** Félix Prado.

**Writing – review & editing:** Aurora Yáñez-Vilches, Alfred C. O. Vertegaal, Félix Prado.

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
