## [Decision Letter · Decision Letter 0]

6 Mar 2024

Dear Dr Prado,

Thank you very much for submitting your Research Article entitled 'Physical interactions between specifically regulated subpopulations of the MCM and RNR complexes prevent genetic instability' to PLOS Genetics.

The manuscript was fully evaluated at the editorial level and by independent peer reviewers. The reviewers appreciated the attention to an important problem, but raised some substantial concerns about the current manuscript. While reviewer 1 was generally positive two other reviewers were not convinced by the support your experiments provided to the claims in the Abstract and in Discussion. More experiments and/or significant modifications of the manuscript may help to convince reviewers and editors. Based on the reviews, we will not be able to accept this version of the manuscript, but we would be willing to review a much-revised version. We cannot, of course, promise publication at that time.

Should you decide to revise the manuscript for further consideration here, your revisions should address the specific points made by each reviewer. We will also require a detailed list of your responses to each review comments and a description of the changes you have made in the manuscript.  Copy of the revised manuscript with tracked or marked up changes would help for the second round of reviews.

If you decide to revise the manuscript for further consideration at PLOS Genetics, please aim to resubmit within the next 60 days, unless it will take extra time to address the concerns of the reviewers, in which case we would appreciate an expected resubmission date by email to plosgenetics@plos.org.

We are sorry that we cannot be more positive about your manuscript at this stage. Please do not hesitate to contact us if you have any concerns or questions.

Yours sincerely,

Dmitry A. Gordenin, Ph.D.

Academic Editor

PLOS Genetics

Gregory P. Copenhaver

Editor-in-Chief

PLOS Genetics

Reviewer's Responses to Questions

**Comments to the Authors:**

Reviewer #1: Review Summary:

Using proteomics, microscopy, genetics and various molecular biology approaches, the authors identified a novel physical interaction between the MCM helicase and ribonucleotide reductase RNR, and characterized the importance of this interaction for preventing genome instability. This manuscript is centered around the themes of spatio-temporal control of complexes and DNA synthesis, and the physical interaction between MCM and RNR involves small subpopulations. These interactions are independent of subcellular localization of the major pools of MCM and RNR in a cell and increase in response to exogenous DNA damage. Partial disruption of this interaction impairs the release of Rad52 from DNA repair centers, but RPA release does not occur. In the absence of the MCM-RNR interaction, Rad52 persists at repair centers and causes genome instability in the form of hypermutation and hyperrecombination.

As MCM was previously shown to interact with Rad51 and Rad52, the authors looked for other MCM interactors in the absence or presence of MMS. These results demonstrate novel roles for MCM and RNR, independent of their well-characterized roles in DNA replication and dNTP synthesis. The authors used the Mcm4-VC construct, which was partially defective in interacting with all RNR components, as a tool to study partial disruption of the MCM-RNR complex. This allowed them to study the biological consequences of loss of the MCM-RNR complex.

The results are original and of importance to researchers in the fields of DNA replication and repair, and these findings will be of broader interest to researchers in the field of genome integrity. The high conservation of the MCM and RNR complexes supports the possibility that that the results presented here using budding yeast as a model system should be of relevance to those studying mammalian systems.

The methodology is rigorous and the authors provide excellent detailed descriptions of the experiments performed and the rationale for each one. The authors also provide substantial evidence for their conclusions.

The manuscript could be strengthened as follows:

There are many concepts described in detail in the Results section that are not introduced in the Introduction. In some sections, this makes it difficult to relate the results back to the original goals of the work and the bigger picture. In my opinion, moving some of the detailed background currently found in the Results to the Introduction section would streamline the manuscript.

It would be helpful to have a model (could be panel in Figure 6) that gives a summary of the physical interactions, cellular localization (with respect to cell cycle stage and DNA damage), and biological roles for MCM, RNR and the MCM/RNR complexes. In some of the sections in the text, the descriptions became a very detailed and it becomes difficult to tie concepts together. A model or graphical abstract that summarizes the localization, subcomplexes, functions of the major complexes as well as those identified here would help to orient readers, especially non-experts.

There are two typos where Rnr3 has been incorrectly written as Rrn3 (page 5 second paragraph and page 18 second paragraph).

The authors measured spontaneous mutagenesis by determining the CAN1R frequency in their various strains in Figures 6E and F. Were any of these CANR mutants sequenced to determine the what mutations are formed following failure of the MCM-RNR interaction? This mutation specificity could provide evidence as to the source of these mutations (TLS, Polymerase delta, etc) and/or the output of repair processes due to excessive Rad52 at DNA repair centers and inefficient PCNA loading (as suggested on page 19 of the Discussion).

The authors should discuss the possible other biological consequences of persistent Rad52 at sites of DNA repair. For example, could this prevent the binding of other proteins? These would be ideas to be explored in future work.

Reviewer #2: This manuscript describes a novel set of interactions between the MCM complex and components/regulators of RNR. Temporal and spatial aspects of one of these interactions is analyzed using Mcm4 and Rnr4 fused to chimeric Venus fluorescent tags. A peculiar situation is observed where expression of the Mcm4-VN fusion protein appears to perturb the RNR interaction. The Rnr4-VN fusion, however, potentiates the interaction. At this point, the authors transition to using Mcm4-VN as a reagent to probe the significance of the RNR interaction, employing the logic that if a Mcm4-VN phenotype is ameliorated by Rnr4-VN it can be attributed to an interaction defect. Such discrimination becomes important, as Mcm4-VN exhibits phenotypes likely arising from perturbing MCM function in DNA replication.

The two primary phenotypes observed with Mcm4-VN that can be rescued (to variable extents) by Rnr4-VC are a delay in the disappearance of Rad52 foci following DNA damage/replication stress and an elevation in mutations attributable to TLS. There is also an ~ 2X increase in dNTP pools that is mildly suppressed by Rnr-VC.

All told, there is a lot of work here with a robust set of data. The first part of the manuscript characterizing the interactions could easily be reworked to stand on its own, albeit without the functional connection. The second part of the paper seems ambitious, painting some interesting yet broad strokes that go beyond the data shown. A main concern, of course, is interpretation of the Mcm4-VN mutant. In addition, the central phenotypes do not necessarily coalesce into a clear functional model.

Specific comments on first part of paper:

1) One question re first part of the manuscript is whether the shifted form of Rnr4 is tied into the observation that, as I followed, Dun1 is required for the interaction with the R2 subunits.

2) As I followed, the only evidence that RNR binds to the MCM hexamer and not just Mcm4 is the supplemental figure with Mcm7. Is that correct?

3) I thought “these results indicate that the Mcm4- VC/Rnr4-VN complex is mobilized from the nucleus to the cytoplasm independently of the individual MCM and R2 complexes” was an overstatement. The data suggest the timing is different, but, for example, this could reflect a differential susceptibility of the fraction to the same form of regulation.

Specific comments on second part of paper:

1) Are Mcm4-VN phenotypes recessive? It seems possible a pool of Mcm4-VC interferes with a function that is separable from the interaction with RNR, but the enhanced interaction provided by Rnr4-VN titrates Mcm4-VC away from this pool. In such a situation, Mcm4-VN might act in a dominant negative manner. Is Rnr3 constitutively upregulated in the Mcm4-VN mutant?

2) Given the phenotypes associated with Mcm4-VN, the authors should comment on whether this introduces caveats in using Mcm4-VN as a physiological reporter for the Rnr4 interaction.

3) The blots shown in Figure 3B need a more comprehensive description in the text or legend. Different antibodies are used to detect the Mcm4 and Rnr4 fusions. Nonetheless, the apparent massive increase in recovery of Rnr4 relative to Mcm4 IP levels is notable. Some regions of the bands, zoomed in, appear pixelated whereas other areas do not. Is this a saturation effect? The bottom RNR panel, and how it relates to the highly shifted form, needs a clearer explanation.

4) The discussion emphasizes a primary role for the MCM/RNR interaction is likely to be (apparently negative) regulation of dNTP levels. The main result is that the Mcm4-VN fusion has persistent Rad52 foci. How those two things are interconnected is at present unclear, giving that part of the analysis a preliminary feel.

5) Similarly, I also did not follow how the previous work on a non-HR role for Rad52 in TLS specifically informs a possible connection between dNTP levels and dispersal of Rad52 from foci. Rightly or wrongly, I was left feeling that a synthesis of ideas is envisioned that is not yet supported by the current data, evocative as it may be.

6) A model figure with what is envisioned by “hub” would contribute to the paper. What would be the advantage of compartmentalizing/complexing the RNR and signaling factors together in this fashion?

Reviewer #3: The study by Yáñez-Vilches and coworkers investigates the interaction between Mcm4 and several subunits of the ribonucleotide reductase complex (Rnr). A proteomic interaction screen with Mcm4-GFP in the presence and absence of low-dose MMS suggests that Dun1 and Rad53 show increased binding with Mcm4, and regulators of Rnr. The volcano plots that are shown would suggest that Rnr3 binding is not significantly enhanced after MMS treatment. The way this is described in the manuscript is ambiguous (page 5). Nevertheless, the authors show by pulldown of Mcm4-GFP that Rnr can be co-immunoprecipitated. The claim that the interaction is enhanced by MMS is only visible on the blot in Figure 1E; the other blots suggest that there is little or no change.

The authors show numerous experiments that alter the cellular localization of Mcm4 and do not find that this affects the localization of Rnr. They conclude that only a small subpopulation of Mcm4 and Rnr subunits bind to each other. They subsequently tag Mcm4 and Rnr4 with the N- and C-terminal parts of the Venus fluorophore, respectively. With this sensitive interaction measure they confirm that small fractions of Mcm4-VC and Rnr4-VN interact in the nucleus in G1 and then gradually accumulate in the cytoplasm in S/G2. Although this is interesting, it remains unclear what the biological significance of this result is. The localization does not significantly change under DNA damage conditions.

Furthermore, the authors establish that Mcm4-VC is compromised in binding untagged Rnr4. Mcm4-VC expression causes low level spontaneous damage that cannot be suppressed by Rnr4-VN expression. The authors then investigated whether the Mcm4 mutation affects double strand break (DSB) repair kinetics. However, Mcm4-VC expressing cells appear to repair DSBs with wildtype kinetics. Only when cells are the authors studied both RAD52 and RPA foci did they see a slight delay in the release of Rad52 from DNA after DSB repair. Release of Rad52 was similar in Mcm4-VC and Mcm4-VC Rnr4-VN strains arguing that the interaction between Mcm4 and Rnr subunits does not have a significant role in repair kinetics.

Because it is unclear why Rad52 release is delayed in Mcm4-VC cells under the experimental conditions the authors use, I find it difficult to draw any strong conclusions from this part of the study. It might have nothing to do with the diminished binding to Rnr.

In summary, I don’t find the conclusions as they are presented in the abstract justified by the data. The data show that there is an interaction between Mcm4 (and Mcm7) and Rnr subunits but whether the interaction that might be indirect or indirect serves any biological function remains elusive.

**Have all data underlying the figures and results presented in the manuscript been provided?**

Large-scale datasets

---

## [Decision Letter · Decision Letter 1]

8 May 2024

Dear Dr Prado,

We are pleased to inform you that your manuscript entitled "Physical interactions between specifically regulated subpopulations of the MCM and RNR complexes prevent genetic instability" has been editorially accepted for publication in PLOS Genetics. Congratulations!

Please note Review #2 identified a small typo (see below) which you can take care of as you prepare your final draft for the production team (the editorial team will not need to re-evaluate).

Yours sincerely,

Dmitry A. Gordenin, Ph.D.

Academic Editor

PLOS Genetics

Gregory P. Copenhaver

Section Editor

PLOS Genetics

Comments from the reviewers (if applicable):

Reviewer's Responses to Questions

**Comments to the Authors:**

Reviewer #1: All of my suggestions to the authors to improve the manuscript were implemented through adjustments and additions to the text as well as Figure 6G. I am satisfied with the revised manuscript.

Reviewer #2: I appreciate the very thorough response of the authors to the reviews. Adding the experiments to test whether the MCM4-VC allele was recessive or acting as a dominant negative was important for the second part of the paper. The evidence indicates it is acting in a recessive manner, a point critical for interpretation of the phenotypes. The changes made to the text and figures also help to share the story in a more linear manner that will make it easier for readers to follow.

There is one typo I found to correct - on pg. 13 of the manuscript version with the changes shown, “A limitation of this approach it .. should read “A limitation of this approach is ...

Reviewer #3: The additional experiments and text edits have improved the manuscript significantly. I still find the evidence for the biological significance of the Mcm-Rnr interaction limited, but the authors have done a good job clarifying their interpretations.

**Have all data underlying the figures and results presented in the manuscript been provided?**

Reviewer #1: Yes

Reviewer #2: Yes

Reviewer #3: Yes

PLOS authors have the option to publish the peer review history of their article (what does this mean?). If published, this will include your full peer review and any attached files.

Reviewer #1: No

Reviewer #2: No

Reviewer #3: **Yes: **Anja Katrin Bielinsky

**Data Deposition**

http://datadryad.org/submit?journalID=pgenetics&manu=PGENETICS-D-24-00082R1

**Press Queries**

---

## [Editor Report · Acceptance letter]

17 May 2024

PGENETICS-D-24-00082R1 

Physical interactions between specifically regulated subpopulations of the MCM and RNR complexes prevent genetic instability 

Dear Dr Prado, 

We are pleased to inform you that your manuscript entitled "Physical interactions between specifically regulated subpopulations of the MCM and RNR complexes prevent genetic instability" has been formally accepted for publication in PLOS Genetics! Your manuscript is now with our production department and you will be notified of the publication date in due course.

With kind regards,

Anita Estes

PLOS Genetics

On behalf of:
